# Structural basis and synergism of ATP and Na$^+$ activation in bacterial K$^+$ uptake system KtrAB

Wesley Tien Chiang [1,14], Yao-Kai Chang [2,14], Wei-Han Hui[3], Shu-Wei Chang[3,4], Chen-Yi Liao[1], Yi-Chuan Chang[1], Chun-Jung Chen [5], Wei-Chen Wang [6], Chien-Chen Lai [6,7], Chun-Hsiung Wang[2], Siou-Ying Luo[2], Ya-Ping Huang[2], Shan-Ho Chou[1], Tzyy-Leng Horng[8], Ming-Hon Hou[9], Stephen P. Muench [10], Ren-Shiang Chen [11], Ming-Daw Tsai [2,12] ✉ & Nien-Jen Hu [1,13] ✉

The K$^+$ uptake system KtrAB is essential for bacterial survival in low K$^+$ environments. The activity of KtrAB is regulated by nucleotides and Na$^+$. Previous studies proposed a putative gating mechanism of KtrB regulated by KtrA upon binding to ATP or ADP. However, how Na$^+$ activates KtrAB and the Na$^+$ binding site remain unknown. Here we present the cryo-EM structures of ATP- and ADP-bound KtrAB from *Bacillus subtilis* (BsKtrAB) both solved at 2.8 Å. A cryo-EM density at the intra-dimer interface of ATP-KtrA was identified as Na$^+$, as supported by X-ray crystallography and ICP-MS. Thermostability assays and functional studies demonstrated that Na$^+$ binding stabilizes the ATP-bound BsKtrAB complex and enhances its K$^+$ flux activity. Comparing ATP- and ADP-BsKtrAB structures suggests that BsKtrB Arg417 and Phe91 serve as a channel gate. The synergism of ATP and Na$^+$ in activating BsKtrAB is likely applicable to Na$^+$-activated K$^+$ channels in central nervous system.

Bacteria can be found in a wide variety of habitats with fluctuating salinity, pH and temperature. Sophisticated machineries are therefore essential for bacteria to achieve an immediate physiological adjustment in order to survive in such variable environments[1]. Osmoadaptation is generally regulated by fine-tuning the concentrations of intracellular potassium ions and osmolytes in bacterial cells[2], where TrkAH and KtrAB systems are responsible for the efficient uptake of K$^+$ ions in response to the osmoshock[3,4]. Both systems share a common protein quaternary structure: transmembrane subunits responsible for K$^+$ permeation, and regulatory subunits located in the cytosol, forming a ring-like structure for association with the transmembrane subunits. The regulatory subunits, TrkA and KtrA, which are referred to as regulator of K$^+$ conductance (RCK) proteins, can bind to signaling molecules in the cytosol and control the gating of the transmembrane pores, TrkH and KtrB, respectively. The KtrAB system is of particular interest due to its variety of binding ligands, such as NADH[5], ATP and

[1]Graduate Institute of Biochemistry, National Chung Hsing University, Taichung 402202, Taiwan. [2]Institute of Biological Chemistry, Academia Sinica, Taipei 115201, Taiwan. [3]Department of Civil Engineering, National Taiwan University, Taipei 106319, Taiwan. [4]Department of Biomedical Engineering, National Taiwan University, Taipei 10663, Taiwan. [5]Life Science Group, Scientific Research Division, National Synchrotron Radiation Research Center, Hsinchu 30092, Taiwan. [6]Institute of Molecular Biology, National Chung Hsing University, Taichung 402202, Taiwan. [7]Graduate Institute of Chinese Medical Science, China Medical University, Taichung 406040, Taiwan. [8]Department of Applied Mathematics, Feng Chia University, Taichung 407102, Taiwan. [9]Institute of Genomics and Bioinformatics, National Chung Hsing University, Taichung 402202, Taiwan. [10]School of Biomedical Sciences, Faculty of Biological Sciences and the Astbury Centre for Structural Molecular Biology, University of Leeds, Leeds LS2 9JT, UK. [11]Department of Life Science, Tunghai University, Taichung 407224, Taiwan. [12]Institute of Biochemical Sciences, National Taiwan University, Taipei 106319, Taiwan. [13]Ph.D Program in Translational Medicine, National Chung Hsing University, Taichung 402202, Taiwan. [14]These authors contributed equally: Wesley Tien Chiang, Yao-Kai Chang ✉e-mail: mdtsai@gate.sinica.edu.tw; njhu@nchu.edu.tw

ADP[6,7], c-di-AMP[8–10], Ca²⁺ and Mg²⁺ [11], and Na⁺ [12–14]. However, the molecular mechanisms of these regulatory ligands toward KtrAB are not fully understood.

The KtrAB system is composed of the transmembrane KtrB dimer, each containing four structurally similar domains (D1-D4) (Fig. 1a and Supplementary Fig. 1) arranged around a four-fold pseudosymmetry axis normal to the cell membrane, and the regulatory KtrA octamer (tetramer of homodimer, Fig. 1a) with a ring-like structure, known as the RCK gating ring. ATP binding to KtrA octamer activates K⁺ flux activity of KtrB, whereas ADP binding inactivates it[6,7,15]. Crystal structures of ATP- and ADP-bound KtrA from *Bacillus subtilis* (BsKtrA) show a four-fold symmetric square-like ring and a two-fold symmetric diamond-like ring, respectively[7]. The crystal structure of BsKtrAB complex demonstrates that the square-like octameric ring of ATP-BsKtrA imposes a steric hindrance on the D1M2b helix of BsKtrB, forming a helical hairpin interacting with the BsKtrA octameric ring[7]. The cryo-EM structure of ADP-bound KtrAB from *Vibrio alginolyticus* (VaKtrAB) indicates that the D1M2b region is extended as a continuous helix penetrating into the diamond-like VaKtrA octameric ring[16]. The extended α-helix of D1M2b restricting the passage of K⁺ was proposed to be a critical structural component in the gating of the VaKtrB pore[16]. Additionally, it was speculated that the intramembrane loop, a 15-amino-acid-long segment in the middle of the D3M2 helix protruding into the pore cavity under the selectivity filter, may also play a critical role in gating[17,18]. However, high-resolution structures of both ATP- and ADP-bound KtrAB complexes, preferably from a single bacterial

species, are required for establishing the ligand-regulated gating mechanism at the atomic level.

Furthermore, Na⁺ ions have been suggested to play a role in the activation of the KtrAB complex[12,19]. In vivo observations implicated a sophisticated gating mechanism using Na⁺ as the activator for K⁺ influx, because high intracellular concentration of Na⁺ disrupted many physiological functions[20], and activation of K⁺ uptake was a critical response to counteract the harmful effects caused by excessive Na⁺. The Na⁺-dependent activation of K⁺ flux was also observed in Na⁺-activated K⁺ ($K_{Na}$) channels, encoded by *Slo2.1* and *Slo2.2*, in the central nervous system[21–23], which are essential to prevent the overload of depolarizing Na⁺. Dysfunction of $K_{Na}$ channels is associated with many neuropathological disorders[24,25]. $K_{Na}$ is a homotetrameric channel covalently linked to the RCK domain. The cryo-EM structures of chicken Slo2.2 at closed and open states have been reported[26,27], and the Na⁺ binding sites of human Slo2.2 have just been determined[28]. Interestingly, the Na⁺-dependent activation of $K_{Na}$ channels can be stimulated by binding of NAD⁺, presumably at the RCK domain[29], implicating a synergy between Na⁺ and nucleotides in activation of K⁺ influx. However, little is known about the mechanism of activation from the structural perspective.

In this study, we uncover the Na⁺-binding site in ATP-bound BsKtrAB using cryo-EM and X-ray crystallography and demonstrate a synergistic activation mechanism of BsKtrAB involving both ATP and Na⁺. We elucidate the structural components responsible for K⁺ gating by comparing the high-resolution cryo-EM structures of ATP- and ADP-

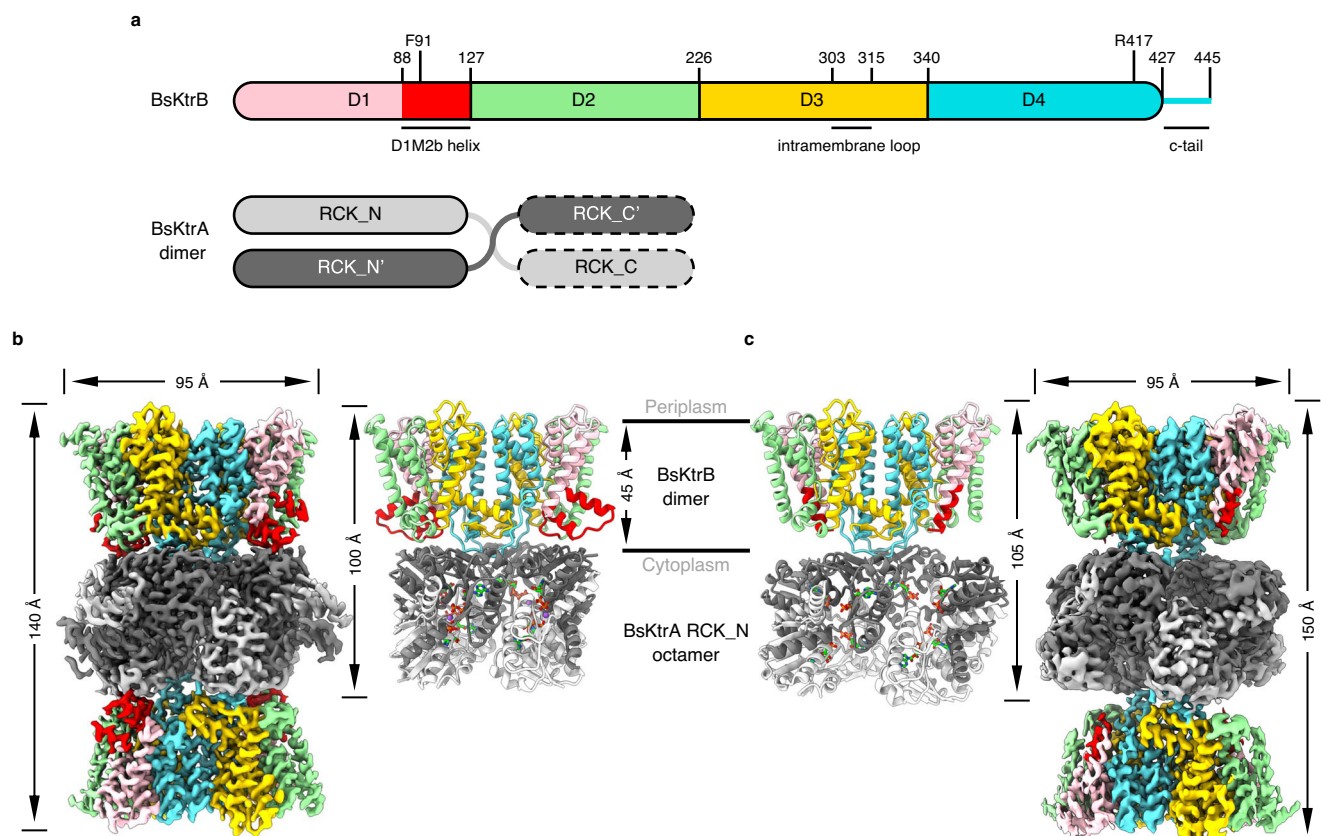

**Fig. 1 | Cryo-EM structures of BsKtrAB complexes. a** The schematic representation of BsKtrB and BsKtrA. BsKtrB is colored by domains with the color scheme: D1 (pink), D2 (green), D3 (yellow) and D4 (cyan). The D1M2b helix in D1 domain of BsKtrB is highlighted in red. The intramembrane loop in D3 domain (303–315) is displayed. The essential residues (F91 and R417) involved in gating are indicated. Each subunit of a BsKtrA homodimer, colored in dark and light gray contains an RCK_N subdomain, a crossover helix and an RCK_C subdomain. The latter is shown

with dashed line because the cryo-EM maps were not clearly resolved. **b, c** Cryo-EM density maps contoured with a threshold of 4.5 σ (showing the KtrB₂A₈B₂ arrangement). The structural models of (**b**) ATP-BsKtrAB (Structure II) and (**c**) ADP-BsKtrAB (structure III) are presented as cartoon and colored with the same color code as in (**a**). The partially built models of the BsKtrA RCK_C subdomains in (**b**) (Structure II) were not shown for clarity. The ATPs in (**b**) and ADPs in (**c**) are shown as lime sticks, and the Na⁺ ions (**b**) are shown as purple spheres.

bound BsKtrAB from a single bacterial species and proposed a comprehensive channel gating mechanism. These findings shed light on the K$^+$ uptake in response to Na$^+$ by coupling the ligand-induced conformational change of the RCK domain to the pore gating, and may provide a mechanistic framework for K$_{Na}$ channels.

## Results

### Na$^+$ binding site in ATP-BsKtrAB and ATP-BsKtrA

As noted in earlier crystallographic studies, an electron density blob was observed in the intra-dimer interface of KtrA in the ATP-BsKtrAB complex, and Mg$^{2+}$ was assigned at this site[7,11]. We first tried to validate the assignment of Mg$^{2+}$ by solving the cryo-EM structures of ATP-BsKtrAB in the previously published buffer condition[2] with the addition of 2 mM Mg$^{2+}$ (Structure I) (Supplementary Fig. 2), and with the Mg$^{2+}$ removed by adding 2 mM EDTA and 1 mM EGTA (Structure II) (Fig. 1b). The datasets were processed and refined to 2.5 Å and 2.8 Å, respectively (Supplementary Fig. 3,4 and Supplementary Table 1). We also solved the cryo-EM structure of ADP-BsKtrAB at the resolution 2.8 Å (Structure III) (Fig. 1c, Supplementary Fig. 5 and Supplementary Table 1). As previously observed, BsKtrAB complexes reveal a nonphysiological quaternary structure of KtrB$_2$A$_8$B$_2$ assembly due to the identical interface for KtrB dimer binding on both sides of KtrA octameric ring[7,15,16]. The models were not completely built in BsKtrA RCK_C subdomain because the cryo-EM density maps were not clearly resolved in this region (Supplementary Table 2). Mg$^{2+}$-added and Mg$^{2+}$-free ATP-BsKtrAB cryo-EM structures (Structure I and Structure II) are virtually identical with an r.m.s.d. of 0.30 Å over 2104 C$_\alpha$ atoms (Supplementary Fig. 6a). Comparing our cryo-EM structures with previously published crystal structures, both ATP-BsKtrAB cryo-EM structures (Structure I and Structure II) are similar to the crystal structure of ATP-BsKtrAB[7] (PDB ID 4J7C) (Supplementary Fig. 6b), while ADP-BsKtrAB cryo-EM structure (Structure III) is structurally comparable to the low-resolution crystals structure of ADP-BsKtrA$_{\Delta C}$B[15] (PDB ID 5BUT) (Supplementary Fig. 6c).

ATP and ADP share the same binding site located at the intra-dimer interface of KtrA RCK_N subdomains[5,7] (Fig. 2a, b), and the ligands can be unambiguously assigned in our cryo-EM structures (Fig. 2c, d). A clear region of unassigned cryo-EM density between the two adjacent ATPs at the intra-dimer interface was observed in Structure I (Supplementary Fig. 7a). Interestingly, the density at this site was still discernable in Structure II, despite the presence of chelators for divalent ions (Fig. 2c, e). The density at the intra-dimer interface of BsKtrA (Structure II) appears to be right on the plane of the horizontal C2 symmetry axis parallel to the membrane, along the BsKtrA dimer interface (Supplementary Fig. 8a). To further confirm the density, we reconstructed a map using the same dataset with a C2 symmetry axis oriented perpendicularly to the membrane bilayer at the BsKtrB dimer interface, and refined at a final resolution of 2.9 Å (Structure IIa) (Supplementary Fig. 8b–e and Supplementary Table 1). Additionally, a map without imposing symmetry (C1) was also reconstructed and refined to 3.0 Å (Structure IIb) (Supplementary Fig. 8f–i and Supplementary Table 1). The maps processed with the vertical C2 symmetry axis or C1 symmetry showed a visible density comparable to the original map with the horizontal C2 symmetry axis (Supplementary Fig. 9). Because the models (Structures II, IIa and IIb) revealed no obvious change in this region, we herein used Structure II for the following structural analysis and discussion due to the better resolution.

As previous studies have demonstrated that the activity of KtrAB is Na$^+$-activated[12,19], and the sample contained 70 mM Na$^+$ in the protein buffer, we suspected that Na$^+$ ions may bind at this site. Structural refinement of Na$^+$ at the site (Structure II) revealed an octahedral coordination geometry with the distances ranging from 2.50 Å to 2.63 Å (Fig. 2e), which are within the range of Na$^+$ coordination distance of 2.13 Å to 2.76 Å[30]. The assigned Na$^+$ is coordinated by the carboxylate oxygens of Glu125 residues and the oxygens in the

γ-phosphates of ATPs from both BsKtrA protomers in the intra-dimer interface. The side chains of Arg16 residues reveal an energetically unfavorable conformation pointing to Na$^+$ in the close vicinity. Nevertheless, the repulsive force would be compensated by the γ-phosphates of both ATPs and the carboxylate groups of both Glu125 residues. It is worth noting that the protein sample also contained 30 mM K$^+$. Refinement of K$^+$ at the assigned site showed an identical coordination geometry with the same coordinating atoms as Na$^+$ with the distances ranging from 2.81 Å to 2.89 Å (Supplementary Fig. 7b).

To scrutinize the unambiguity of the cation at the binding site, we co-crystallized BsKtrA with 2 mM thallium acetate in the presence of 150 mM K$^+$ and performed anomalous scattering experiments at its absorption edge (0.975 Å, 12,712 eV). Tl$^+$ provides a good mimic for K$^+$ due to a similar ionic radius[31]. However, anomalous scattering of Tl$^+$ was also utilized to identify Na$^+$-binding sites in the glutamate transporter Glt$_{Ph}$ and the Na$^+$/H$^+$ antiporter PaNhaP[32,33]. Surprisingly, the anomalous density peak representing Tl$^+$ is localized at the same site of the intra-dimer interface (Fig. 2f and Supplementary Table 3). We then performed a competition test by replacing K$^+$ in the BsKtrA protein buffer with 150 mM of Na$^+$. The anomalous peak was diminished with a clear density peak at the site in the F$_o$-F$_c$ omit map (Fig. 2g and Supplementary Table 3), supporting that Na$^+$ is the preferred ion to occupy the binding site at the intra-dimer interface of ATP-BsKtrA.

The ICP-MS analysis demonstrated that the Na$^+$ content of ATP-BsKtrA is higher than apo-BsKtrA, further indicating that Na$^+$ binding to BsKtrA is ATP-dependent (Supplementary Table 4). The molar ratio of specific ATP-associated Na$^+$ to BsKtrA is 0.48:1, approximately in agreement with the stoichiometric relationship of one Na$^+$ ion in an ATP-BsKtrA dimer. Furthermore, K$^+$ content was lower in either apo- or ATP-BsKtrA and no significant difference was shown between the two samples, suggesting that the site favors the binding of Na$^+$.

### Synergistic effects of Na$^+$ and ATP on BsKtrA

To investigate the possible functions of Na$^+$ binding to BsKtrA, we systematically characterized protein stability toward Na$^+$ and other cations. Based on the structural information, the cation could stabilize the ATP binding and thus the complex. We first investigated the stability of ATP-BsKtrA in the presence of Na$^+$ using urea-induced unfolding by monitoring the intrinsic tryptophan fluorescence[34]. BsKtrA was prepared in K$^+$ Buffer and Na$^+$ Buffer (containing 150 mM of either cation) with the titration against Na$^+$ and K$^+$, respectively (see Methods for details). The midpoint urea unfolding concentration (C$_m$) of ATP-BsKtrA was found to be increased in a [Na]-dependent manner, in contrast to little or no effect with apo-BsKtrA and ADP-BsKtrA against Na$^+$ (Fig. 3a, Supplementary Fig. 10). It is noted that ATP-BsKtrA in K$^+$ Buffer revealed a C$_m$ at ~2 M (Fig. 3a, 0 mM Na$^+$ titration), but intriguingly, while prepared in Na$^+$ Buffer, ATP-BsKtrA showed a much higher C$_m$ at ~4.5 M (Fig. 3b, 0 mM K$^+$ titration) and titration against K$^+$ caused little effects on C$_m$, implicating ATP-BsKtrA in the absence of Na$^+$ (K$^+$ Buffer) is more conformationally unstable, and, furthermore, K$^+$ cannot compete in the binding site of ATP-BsKtrA against Na$^+$.

To individually characterize the effects of Na$^+$ and K$^+$ on the thermostability of ATP-bound BsKtrA, we prepared protein samples in Choline Buffer and performed differential scanning fluorimetry (DSF)[35]. The T$_m$ of ATP-BsKtrA showed a [Na$^+$]-dependent increase from 39 °C in the absence of Na$^+$ to 57 °C at 100 mM Na$^+$. The apparent Na$^+$-binding affinity (K$_{dapp}$) was determined to be 27.5 ± 1.6 mM, while no thermostability effect was observed for K$^+$ ions on ATP-BsKtrA (Fig. 3c and Supplementary Fig. 11a). Furthermore, both Na$^+$ and K$^+$ revealed no effect on apo-BsKtrA and ADP-BsKtrA (Supplementary Fig. 11c–f). Substitution of BsKtrA Glu125, which is involved in the coordination with Na$^+$ (Fig. 2), with glutamine (E125Q) abolished the [Na$^+$]-dependent increase of T$_m$ while BsKtrA E125Q was pretreated with ATP (Fig. 3d and Supplementary Fig. 11b), indicating the stabilizing effect of Na$^+$ on ATP-BsKtrA.

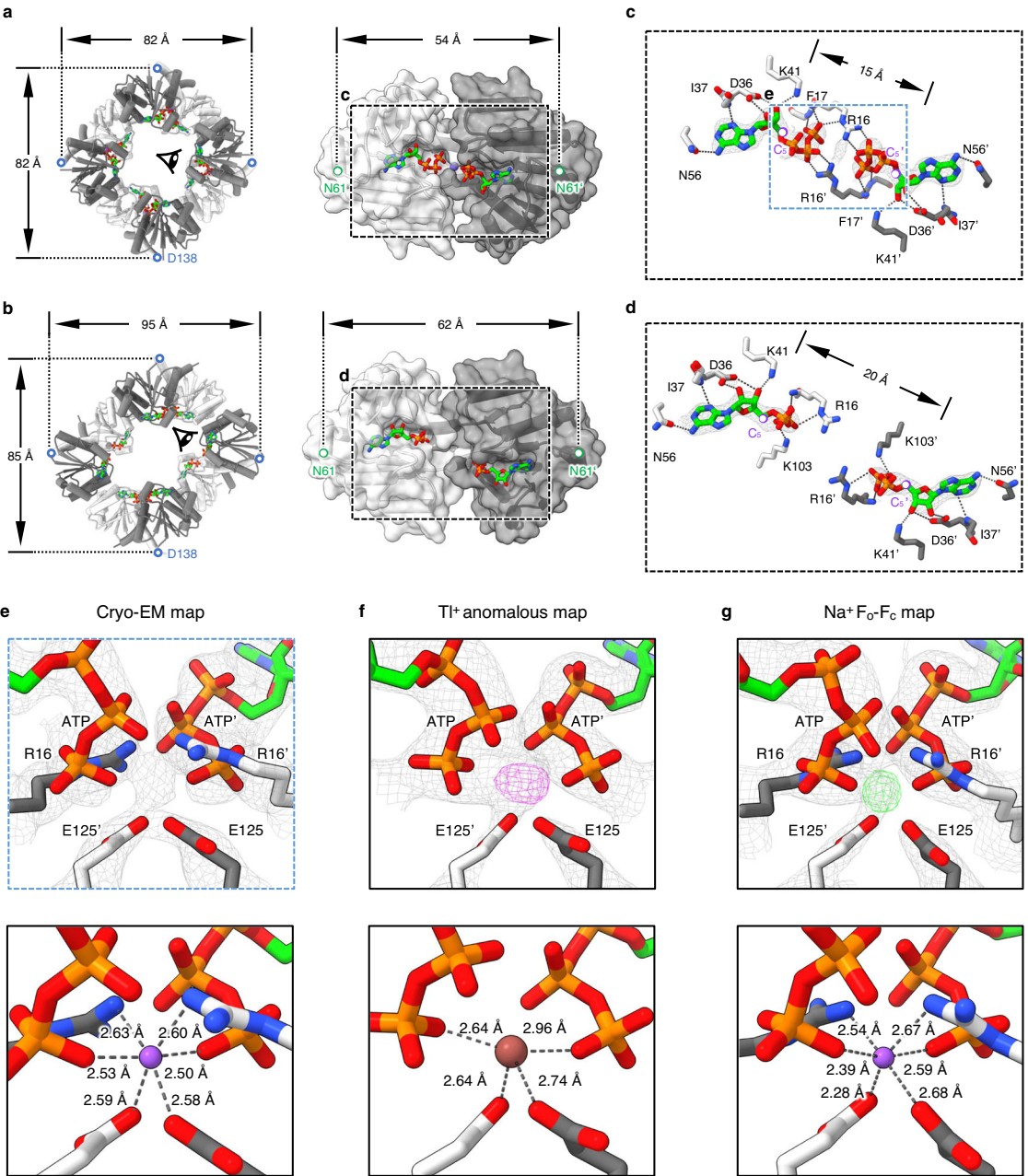

**Fig. 2 | The sodium-binding site of ATP-BsKtrA. a, b** Left panels, BsKtrA octameric rings in cylinder representation at (**a**) ATP- and (**b**) ADP-bound states from ATP-BsKtrAB (Structure II) and ADP-BsKtrAB (Structure III), respectively, with ATP and ADP shown in stick representation. The protomers of each BsKtrA dimer are colored in light and dark gray with semi-transparent surface and cartoon backbone representation. The RCK_C subdomains are omitted for clarity. Right panels, close-up views of (**a**) ATP- and (**b**) ADP-BsKtrA dimers from the perspective as indicated by the eye symbols shown in the left panels. The $C_\alpha$ atoms of Asp138 and Asn61 are shown as blue and lime dots, respectively, with the double-head arrows indicating the distances between designated atoms. **c, d** Close-up views of the detailed (**c**) ATP- and (**d**) ADP-binding sites at the intra-dimer interface of BsKtrA magnified from the dashed boxes defined in (**a**) and (**b**), respectively. The cryo-EM densities of the nucleotides are contoured at 10 σ. The ribose $C_5$ atoms of the two bound (**c**) ATP or (**d**) ADP molecules are shown in purple dots with the respective distances

indicated. **e–g** Upper panels, close-up views of the intra-dimer interface of ATP-BsKtrA from (**e**) the cryo-EM structure (Structure II), (**f**) the crystal structure of $Tl^+$ pretreated ATP-BsKtrA in the presence of 150 mM KCl, and (**g**) the crystal structure of $Tl^+$ pretreated ATP-BsKtrA in the presence of 150 mM $Na^+$. The gray mesh represents the cryo-EM density map contoured at 7.0 σ in (**e**) and $2F_o$-$F_c$ electron density maps contoured at 2.3 σ in (**f**) and 2.0 σ in (**g**). The coordinating amino acid side chains and ATP γ-phosphates are shown in stick representation. The anomalous difference density map of $Tl^+$ (magenta mesh) in (**f**) is contoured at 5.0 σ, but the anomalous difference density cannot be observed in (**g**) even at the contoured level of 4.0 σ. The $F_o$-$F_c$ omit map (green mesh) in (**g**) is contoured at 5.0 σ. Lower panels, coordination geometries of $Na^+$ (purple spheres) in (**e**) and (**g**) and $Tl^+$ (brown sphere) in (**f**) after structure refinement are depicted as dashed lines with distances indicated.

Half-life analysis of BsKtrA incubated at 40 °C also produced results in line with urea unfolding and thermostability assays: in the absence of $Na^+$, ATP alone binding to BsKtrA diminished the protein stability ($t_{1/2} = 9.8 \pm 1.4$ min) compared to the $t_{1/2}$ of apo-BsKtrA ($16.6 \pm 0.8$ min), but simultaneously addition of $Na^+$ and ATP

synergistically increased the half-life of BsKtrA ($t_{1/2} = 70 \pm 2.5$ min) (Fig. 3e and Supplementary Fig. 12). Interestingly, BsKtrA to ATP binding affinities characterized using isothermal titration calorimetry (ITC) revealed comparable dissociation constants in the presence and absence of 200 mM $Na^+$ ($K_d = 1.7 \pm 0.3$ μM and $5.5 \pm 1.0$ μM,

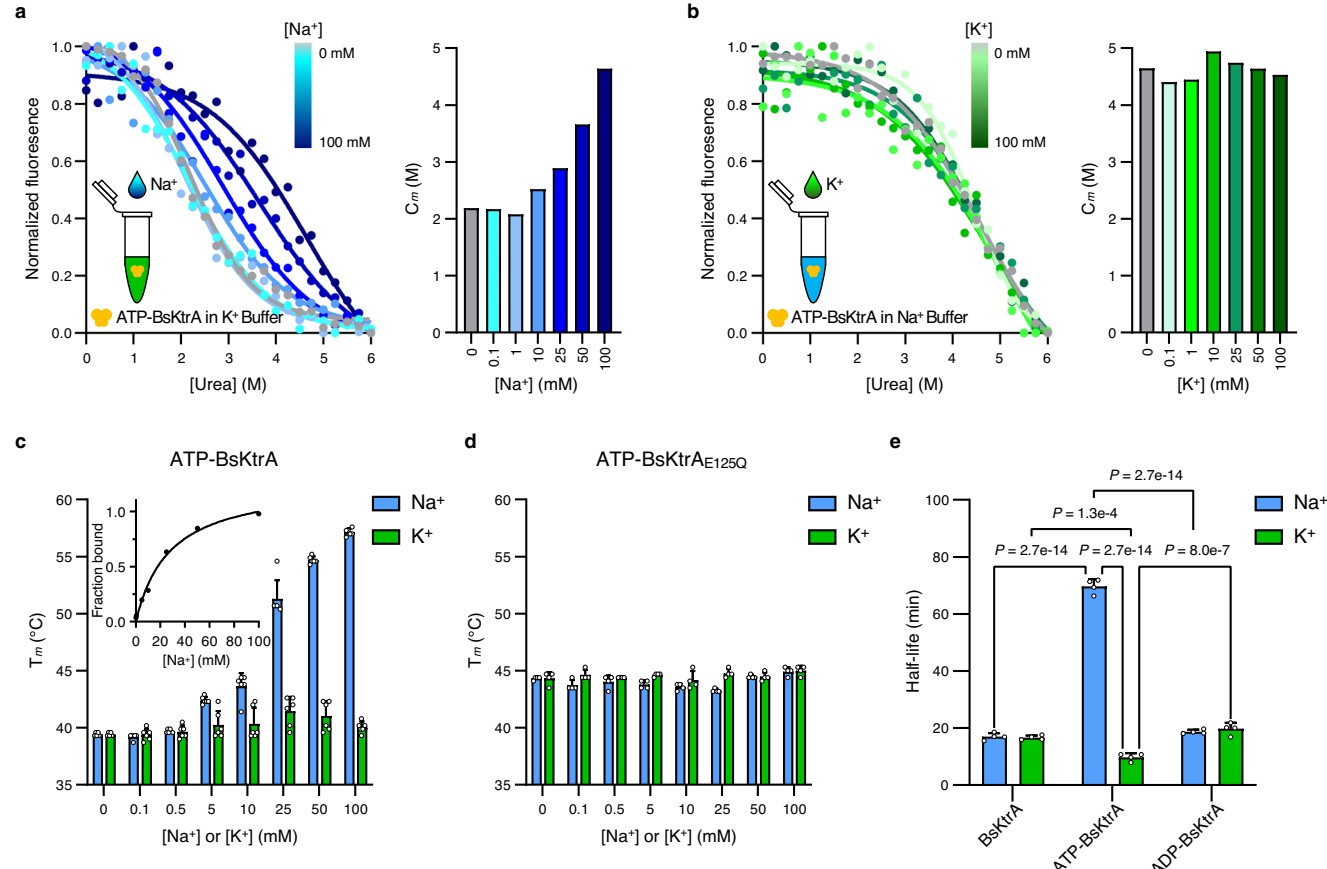

**Fig. 3 | Na⁺ is critical for ATP-BsKtrA stability. a, b** Left panels: intrinsic trypto-phan fluorescence-based urea unfolding assays of ATP-BsKtrA in (**a**) K⁺ Buffer titrated against NaCl and in (**b**) Na⁺ Buffer titrated against KCl, respectively. Right panels: the $C_m$ values plotted against [Na⁺] and [K⁺], respectively. The $C_m$ values were calculated by fitting the data to a sigmoidal 4PL model using GraphPad Prism. Refer to Supplementary Fig. 10a, b for original fluorescence data. **c, d** Differential scanning fluorimetry (DSF) assays demonstrating the $T_m$ of (**c**) ATP-BsKtrA and (**d**) ATP-BsKtrA_{E125Q} titrated with NaCl (blue bars) and KCl (green bars). Data represent the mean ± s.d. with (**c**) $n = 6$ and (**d**) $n = 4$ independent experimental replicates.

Inset **c**, the apparent dissociation constant ($K_{dapp}$) of Na⁺ binding with ATP-BsKtrA. The $K_{dapp}$ was analyzed by fitting the fraction bound derived from the DSF assays against the concentrations of NaCl to a one-site model using GraphPad Prism. Refer to Supplementary Fig. 11a, b for raw data. **e** Half-life constants of BsKtrA in the presence of Na⁺ (blue bars) and K⁺ (green bars). The half-life constants were determined by GraphPad Prism using exponential one-phase decay model. Data represent the mean ± s.d.; $n = 4$ independent experimental replicates. Statistical analyses were performed using two-way ANOVA. Refer to Supplementary Fig. 12 for raw data. Source data for **a**–**e** are provided as a Source Data file.

respectively) (Supplementary Fig. 13), suggesting BsKtrA is capable of binding ATP even in the absence of Na⁺, although this particular ligand-binding state is actually unstable.

Altogether, these biophysical studies indicate that binding of ATP alone to BsKtrA is thermodynamically unstable, while Na⁺ stabilizes BsKtrA in an ATP-dependent manner. The results are in great agree-ment with the structural finding that the two tethered ATP molecules at the intra-dimer interface of BsKtrA is energetically unfavorable; however, the electrostatic repulsion can be stabilized by Na⁺.

### Ca²⁺ binding to BsKtrA

The protein stability and ligand affinity assays described above provide solid evidence that Na⁺ plays a critical role in stabilizing ATP-BsKtrA. These results in the roles of Na⁺ are not necessarily in conflict with the roles of Ca²⁺ and Mg²⁺ reported previously[11], which may also bind to the same Na⁺ site or to different sites. To examine whether divalent cations could interact with BsKtrA in an ATP-dependent manner as previously implicated, we also characterized the thermostability of BsKtrA in the presence of Ca²⁺ or Mg²⁺ (Supplementary Fig. 14a–e). Both Ca²⁺ and Mg²⁺ had no effect on ADP-BsKtrA and only a modest destabilizing effect on apo-BsKtrA (Supplementary Fig. 14a, b, d, e). Interestingly, Ca²⁺ increased the $T_m$ of ATP-BsKtrA in a concentration-dependent manner with an apparent Ca²⁺-binding affinity ($K_{dapp}$) of 207 ± 29 μM

(Supplementary Fig. 14d, f), and the [Ca²⁺]-dependent increase of $T_m$ was abolished in ATP-BsKtrA_{E125Q} (Supplementary Fig. 14g, h), but Mg²⁺ caused little impact on the thermostability of ATP-BsKtrA (Supple-mentary Fig. 14c, e). The ICP-MS analysis indicated that Ca²⁺ content is higher in ATP-BsKtrA (2.529 ± 0.180 μg/ml) than in apo-BsKtrA (N.D.), further substantiating that the Na⁺ binding site is also favorable for binding of Ca²⁺. However, Mg²⁺ contents in both apo- or ATP-BsKtrA are at low level (0.113 ± 0.003 and 0.135 ± 0.004 μg/ml, respectively), and no Mg²⁺-dependent difference is observed between the two sam-ples (Supplementary Table 4). Taken together, the results are in partial agreement with the recent study, suggesting that both Ca²⁺ and Mg²⁺ can bind to this site[11]. Notably, in that published results, Mg²⁺-induced activation of BsKtrAB was not exclusively ATP-dependent, and not even BsKtrA-dependent[11]. Further characterization is necessary to define the regulatory mechanism of divalent cations in KtrAB.

### Na⁺ facilitates ATP-BsKtrAB assembly and elevates the K⁺ flux

The structural data and stability studies presented above suggest that Na⁺ binding stabilizes the conformation ATP-BsKtrA, which could propagate to the BsKtrB to allow a stable quaternary structure of BsKtrAB with fully activated K⁺ flux activity. To test this hypothesis, we first investigated the effects of Na⁺ binding on BsKtrAB assembly using size exclusion chromatography (SEC). ATP-BsKtrA and BsKtrB mixture

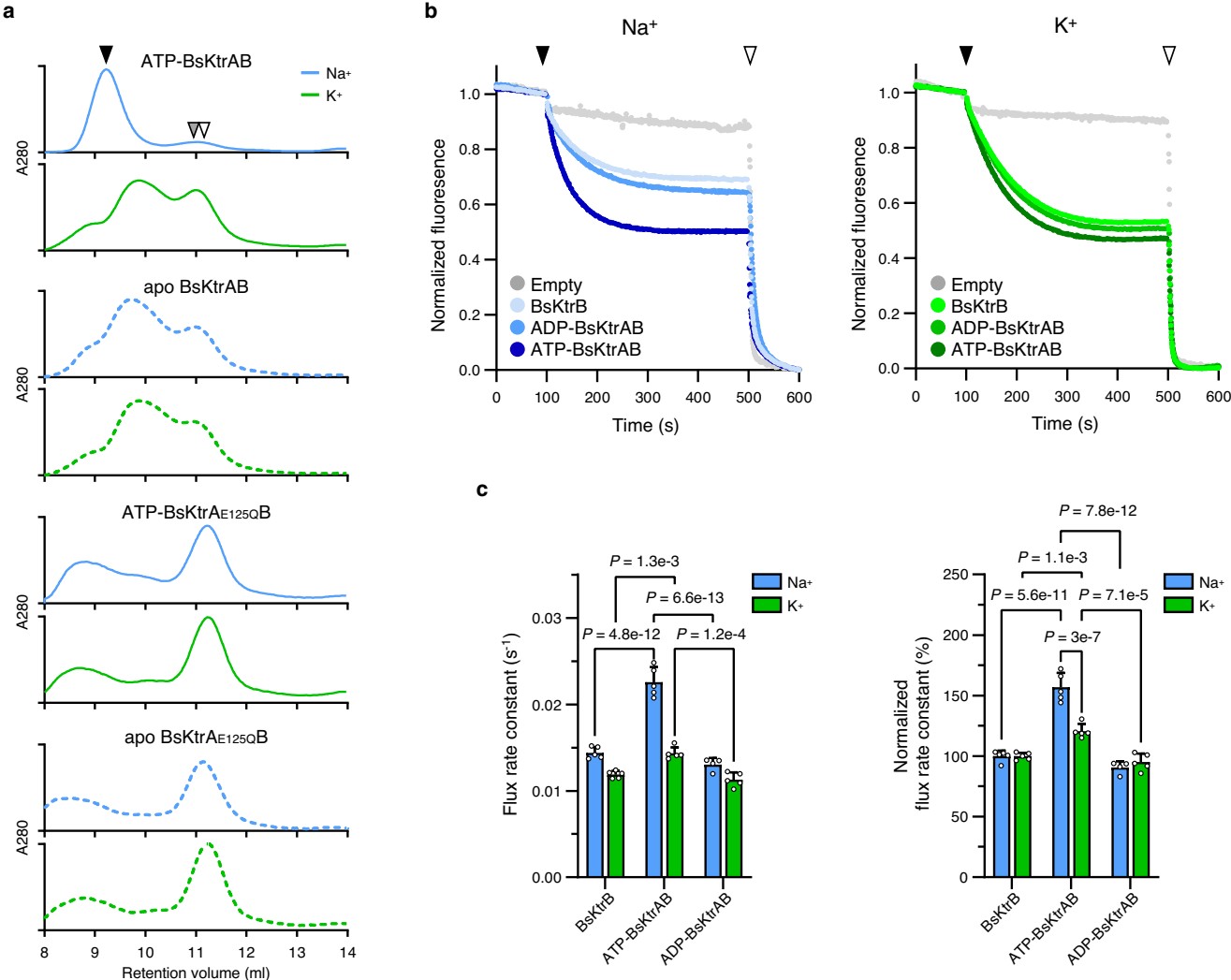

**Fig. 4 | Na⁺ is important for functional assembly of ATP-BsKtrAB. a** The SEC profiles of BsKtrA (apo BsKtrA, ATP-BsKtrA, apo BsKtrA$_{E125Q}$ or ATP-BsKtrA$_{E125Q}$) mixed with BsKtrB in Na⁺ Buffer or K⁺ Buffer with 0.03% DDM. The retention volume for the BsKtrAB complex is benchmarked as the black arrow, while the retention volumes for BsKtrA octamer and BsKtrB dimer are benchmarked as the gray and white arrows, respectively. **b** Fluorescence-based K⁺ flux assays using BsKtrB, ADP-BsKtrAB, and ATP-BsKtrAB in Swelling Na⁺ Buffer (Na⁺, left panel) or Swelling K⁺ Buffer (K⁺, right panel). The addition of H⁺ ionophore CCCP and K⁺ ionophore valinomycin are indicated as black and white arrows. K⁺ flux rate constants were calculated by fitting the data (100–500 s) to a one-phase decay model using GraphPad Prism. **c** K⁺ flux rate constants (left panel) and normalized flux rate constants (right panel) of BsKtrB, ATP-BsKtrAB and ADP-BsKtrAB in the presence of Na⁺ (blue bars) and K⁺ (green bars). The normalized K⁺ flux rate constants were calculated using the respective K⁺ flux rates of BsKtrB in the presence of either Na⁺ or K⁺ as 100%. Data represent mean ± s.d. with $n = 4$ (for ADP-BsKtrAB in Na⁺) or $n = 5$ (for the others) independent experimental replicates. Statistical analyses were performed using two-way ANOVA. The schematic illustration of normalization and analysis are shown in Supplementary Fig. 15. Source data for **a**–**c** are provided as a Source Data file.

in the presence of Na⁺ revealed a uniform species (Fig. 4a, black arrow), representing a stable assembly of ATP-BsKtrAB complex as demonstrated by the cryo-EM structure (Structure II) collected from the indicated fraction. Na⁺ failed to produce a single species of ATP-BsKtrA$_{E125Q}$B assembly (Fig. 4a). In the absence of Na⁺, ATP-BsKtrA may still interact with BsKtrB as shown in Fig. 4a (ATP-BsKtrAB in K⁺ Buffer with 0.03% DDM), but the mixture cannot form a monodisperse species. These results highlight the importance of Na⁺ in ATP-BsKtrAB complex assembly: without the neutralizing effect of Na⁺ at the middle of the two tethered ATP molecules in the BsKtrA dimer, the octameric ring at an energetically unstable state may reveal a stochastic conformation, deteriorating the proper assembly of ATP-BsKtrAB.

Next, to correlate the protein assembly to the functionality, the liposomal K⁺ flux assay was performed to validate the impact of Na⁺ in the synergistic regulation of ATP-activation in the BsKtrAB complex. The K⁺ flux activities of ADP-BsKtrAB showed inconsiderable

difference in the presence of Na⁺ or K⁺, suggesting no effect of either cation in the ADP-bound state. On the other hand, binding of ATP to BsKtrAB enhanced the K⁺ flux rate by 20% in the absence of Na⁺ (Swelling K⁺ Buffer) and the activity was further elevated by 40% in the presence of Na⁺ (Fig. 4b, c). If Glu125, of which the side chain is involved in the coordination of Na⁺ ions (Fig. 2), was replaced by glutamine, the Na⁺-dependent activation was no longer observed (Supplementary Fig. 16). Altogether, the results suggest that Na⁺ binding at the BsKtrA intra-dimer interface coordinated by the γ-phosphates of ATPs and the carboxylate groups of Glu125 stabilizes the square-shaped ATP-BsKtrA octameric ring and maintains the pore-open conformation of BsKtrB (see below), leading to the activated state of BsKtrAB.

**Comparison of ATP- and ADP-BsKtrAB**

Having illustrated the detailed structure of ATP-BsKtrAB and established the structural basis of its activation by synergistic binding of ATP

and Na$^+$, we are in a strong position to examine the different structural effects of ATP and ADP as a way to further understand the gating mechanism of BsKtrAB. To minimize the uncertain effects that Mg$^{2+}$ may cause on the structure of BsKtrAB (Structure I), we hereafter used Mg$^{2+}$-free ATP-BsKtrAB (Structure II) in comparison with ADP-BsKtrAB (Structure III) for the following structural analysis.

Our high-resolution structures of ATP- and ADP-BsKtrAB complexes with clear EM density of BsKtrB in the transmembrane region (Supplementary Fig. 17) allowed a reliable comparison between the two structures, showing distinct conformational rearrangements in both BsKtrA and BsKtrB. First, ADP binding to BsKtrA results in opening of the intra-dimer interface, as observed in the previous study[7], making Na$^+$ binding at this site unfavorable (Fig. 2d). In support, the biophysical data explains why the activating effects of Na$^+$ on BsKtrA are not significant at the ADP-bound state (Fig. 3e and Supplementary Fig. 11e). The conformational change at the intra-dimer interface causes a dimer-to-dimer rotation, rendering the square- and diamond-shaped octameric rings. Second, in the ATP-BsKtrAB structure, the square-shaped ATP-BsKtrA octameric ring causes a steric hindrance for the D1M2b helix of BsKtrB, resulting in a helical hairpin conformation. The helical hairpin offers a contact interface with BsKtrB, which was previously denoted as the tip contact[7] (Supplementary Fig. 18a). In the ADP-BsKtrAB structure, the long axis of the diamond-shaped ADP-BsKtrA octameric ring provides the space to accommodate the elongated D1M2b helices of the two BsKtrB protomers (Supplementary Fig. 18b), although the density of the very C-terminal end of D1M2b helix (Gly103 to Gly124) is poor, likely due to high flexibility in this region. The conformational change is in general agreement with the cryo-EM structure of the ADP-VaKtrAB complex, where the two extended D1M2b helices penetrate into the octameric ring[16]. Albeit the overall conformational changes in ATP- and ADP-bound states are similar to the previous structural data, the high-resolution cryo-EM structures provide deeper insights into the gating mechanism.

### BsKtrB Arg417 and Phe91 control the gate

Compared to ATP-BsKtrAB, the D4M2 of BsKtrB remains a discontinuous helix in ADP-BsKtrAB (Supplementary Fig. 19a), unlike the extended and folded conformation observed in ADP-VaKtrAB[16]. The study of ADP-VaKtrAB proposed that the extended D4M2 helix along with a reorientation of the highly conserved arginine (Arg427) in the middle of D4M2 helix narrows the pore and closes the gate[16]. Interestingly, the equivalent residue Arg417 in BsKtrB is located below the selectivity filter with its side chain pointing to the pore, spatially close to the intramembrane loop and the kink in D1M2b helix[7]. To further examine the gating mechanism in greater detail, we investigated the detailed conformational changes in this region. Comparing the pore surfaces of BsKtrB in ATP- and ADP-bound states, it shows no obvious difference at the selectivity filter (Fig. 5a). The pore radius analysis reveals the most constricted point near the intramembrane loop in the ADP-bound state (Fig. 5b). Surprisingly, the intramembrane loop in both ATP- and ADP-BsKtrAB shows no pronounced open or closed conformational change (Fig. 5c, d and Supplementary Fig. 19a). The intramembrane loop displaying no secondary structures is unexpectedly stable in both ATP- and ADP-bound BsKtrAB complex structures, as shown by the highly discernable cryo-EM density map (Supplementary Fig. 19b). The loop contains a number of glycine residues (Gly304, Gly306 and Gly313), rendering the plasticity of torsion angle to afford the loop structure. A number of polar residues in the intramembrane loop, which are also well conserved in VaKtrB and other bacterial orthologues (Supplementary Fig. 20), exhibit hydrogen bonding interactions through the side chains with the neighboring residues in D2 domain (Fig. 5c, d). The intramembrane loop is thus stabilized beneath the selectivity filter with a specific hydrogen bond network, and the aforementioned polar interactions remain nearly unchanged in ATP- and ADP-bound states (Fig. 5c, d).

On the other hand, the side chain of BsKtrB Arg417, which was speculated to be a gating residue in the middle of D4M2 helix as mentioned above[17,36], forms a hydrogen bond with the carbonyl oxygen of Thr310 in the intramembrane loop of ADP-BsKtrAB, providing a positively charged barrier that obstructs the passage of K$^+$ ions (Fig. 5d). However, the polar interaction between Arg417 and Thr310 is interrupted in the ATP-BsKtrAB structure, where the side chain of Arg417 forms a hydrogen bond with the exposed carbonyl oxygen of Gly87 located at the kink of the discontinuous D1M2 helix (Fig. 5c). In the ADP-BsKtrAB state, due to the fully extended and continuous helix conformation, Gly87 backbone oxygen is engaged in an intrahelical hydrogen bonding in D1M2 helix, hampering the polar interaction with Arg417. The hypothesis of the transition in hydrogen bonding was further supported by the results of MD simulation (Fig. 5e). Notably, in ADP-BsKtrAB, the phenyl ring of Phe91 in D1M2b points toward the pore directly below the hydrogen bond between Arg417 and Thr310 (Fig. 5d), serving as a hydrophobic gate. In ATP-BsKtrAB, the bulky side chain of Phe91 flips away from the pore due to the formation of a helical hairpin (Fig. 5c). Sequence alignment indicates that Arg417 is highly conserved, and the equivalent residue of Phe91 in other bacterial orthologues is either phenylalanine or hydrophobic residues (Supplementary Fig. 20). Arg417 and Phe91 form the narrowest region in the pore at the ADP-bound state, and this region is widened at the ATP-bound state (Fig. 5b), suggesting that these two residues play a pivotal role in the gating mechanism of BsKtrB in response to the ligand-induced conformation changes of BsKtrA octameric ring. Substituting the Phe91 with alanine enhanced the K$^+$ flux rate in the presence of ADP, showing a similar flux rate as ATP (Fig. 5f); however, substitution of Arg417 resulted in protein aggregation, supporting its importance in protein stability. The experimental results were further substantiated by the SMD simulation, where the force profile indicates a free energy barrier at the region of Arg417 and Phe91 in the ADP-bound state (Fig. 5g).

## Discussion

The crystallographic and electron microscopic data of BsKtrAB, together with the ligand binding characterization and activity analysis in this study provide comprehensive structural and functional evidence to deduce a detailed mechanism of the ligand-gated K$^+$ channel (Fig. 6). Furthermore, the structures of the ATP- and ADP-KtrAB complexes are both derived from *B. subtilis*, making the mechanistic analysis more reliable.

As known from previous structural studies[7,16], ATP-binding to BsKtrA causes a square-shaped octameric ring, rendering a steric hindrance to BsKtrB D1M2b helix and thus inducing a helix hairpin conformation, which leads to an expansion in the pore. ADP-binding to BsKtrA induces a diamond-shaped ring, providing adequate space to allow D1M2b helix to be fully extended and thus restricting the pore. By analyzing the high-resolution cryo-EM structures in this study, we demonstrate that Arg417 and Phe91 serve as a gate blocking the K$^+$ flux in the closed state. The structural transition of D1M2b helix into hairpin alters the hydrogen bond interactions near the gate, resulting in the side chain conformational change of Arg417 and Phe91 in the open state. However, in ATP-bound state, BsKtrA octameric ring is intrinsically unstable because charge repulsion between the two tethered ATP molecules in BsKtrA dimer. Consequently, ATP-BsKtrA octameric ring may not be able to efficiently form a functionally active assembly of BsKtrAB complex. Na$^+$ compensates the negative charge and stabilizes ATP-BsKtrA thermodynamically, which in turn facilitates the interaction between ATP-BsKtrA and BsKtrB, and secures the complex assembly specifically at the functionally active state. Nevertheless, the synergism and gating mechanism proposed here are concluded on the basis of protein samples prepared in detergent micelles. Further structural studies using protein samples in lipid membranes are required to substantiate the mechanistic model.

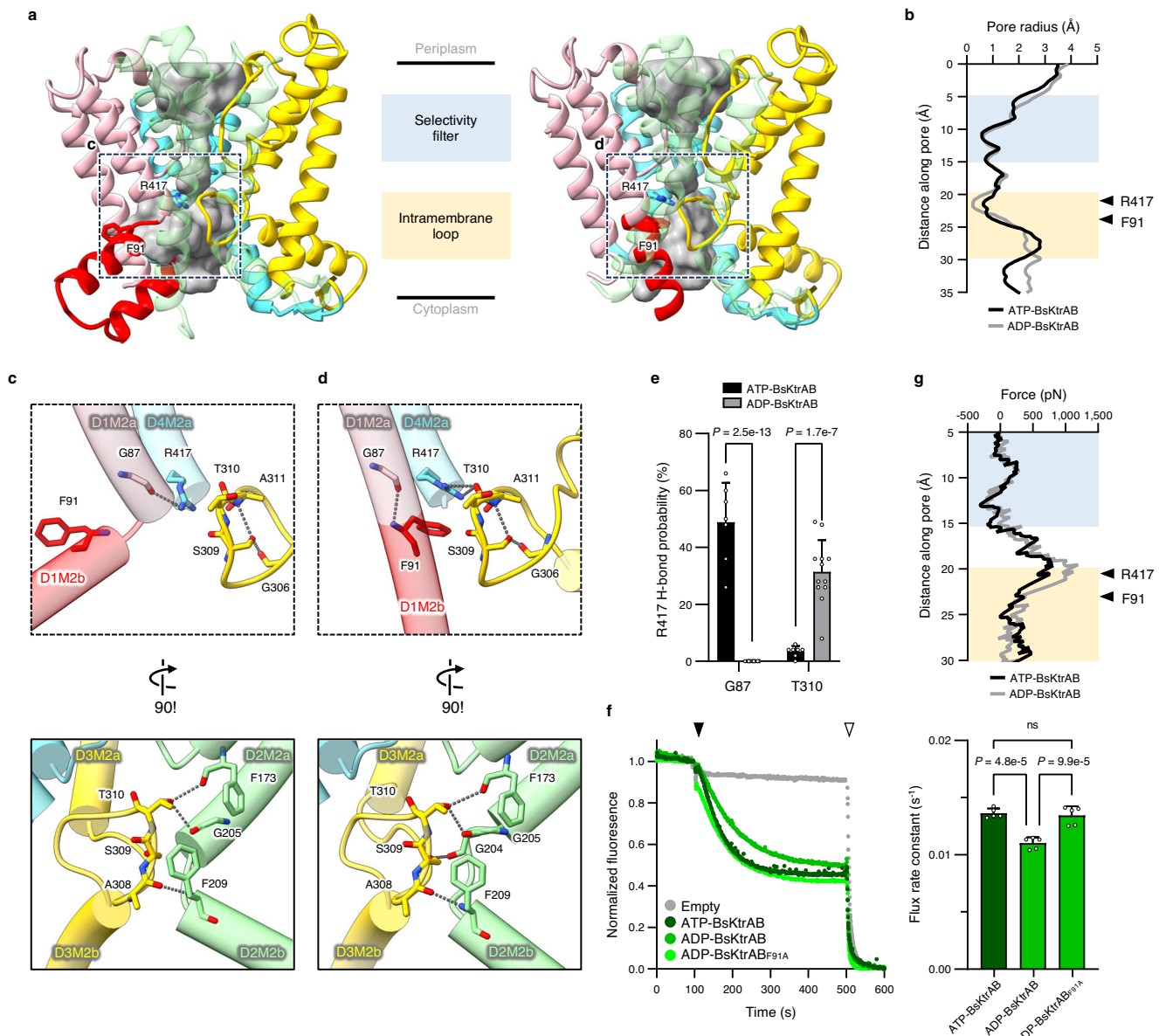

**Fig. 5 | Structure analysis of BsKtrAB. a** Surface representation (gray) of BsKtrB pore in the ATP- (Structure II, left panel) and ADP-bound (Structure III, right panel) BsKtrAB complexes, calculated by HOLLOW. D1-D4 domains and D1M2b helix are highlighted with the same color code as in Fig. 1a. The D2 domain of BsKtrB is transparent for clarity. The side chains of Phe91 and Arg417 are shown in stick representation. **b** Pore radius as a function of the distance along the axis of BsKtrB pore, calculated by HOLE, is plotted with respect to the positions of selectivity filter and intramembrane loop shown in **a**. **c**, **d** A close-up view of the hydrogen bond interactions in the vicinity of the intramembrane loop and D1M2b helix of BsKtrB at (**c**) ATP- and (**d**) ADP-bound states as magnified from the dashed boxes defined in **a**. The side chains of the residues involved in the hydrogen bond interactions (gray dashed lines) are shown in stick representation. **e** The hydrogen bonding probabilities of Arg417-Gly87 and Arg417-Thr310 in the presence of ATP (black) and ADP (gray) are plotted on the basis of molecular dynamics simulation. Data represent the mean ± s.d. with $n = 7$ (ATP) and $n = 12$ (ADP) independent experimental replicates. Statistical analyses were performed using two-way ANOVA. **f** Left panel, fluorescence-based K⁺ flux assays using wild-type BsKtrAB and KtrAB$_{F91A}$ in the presence of ATP or ADP. The addition of H⁺ ionophore CCCP and valinomycin are indicated as black and white arrows. Right panel, plot of flux rate constants, which are calculated by fitting the data (100−500 s) to a one-phase decay model. Data represent the mean ± s.d. with $n = 5$ independent experimental replicates. Statistical analyses were performed by one-way ANOVA, and n.s. indicates no significance ($p \geq 0.05$). **g** The force required for K⁺ to permeate cross the pore as a function of the distance along the pore analyzed using SMD simulation. Source data for (**b**, **e**, **g**, **f**) are provided as a Source Data file.

The TrkHA complex, a member of SKT family consisting of the double-pore channel TrkH and the RCK tetrameric ring TrkA, displays similar signaling responses, gating elements and quaternary structures as those found in KtrAB[36,37]. In the TrkHA system, ATP binding results in a conformational change of TrkA, which triggers a movement of the intramembrane loop to the intracellular side, leading to an open channel conformation[38]. In the BsKtrAB complex, ATP binding does not cause a significant movement of the intramembrane loop to induce

an open conformation. Another feature of BsKtrB differentiating itself from TrkH is its long C-terminal tail snaking into the cytoplasmic pore of the neighboring BsKtrB protomer. The carboxylate group of the very C-terminal residue (Gly445) interacts specifically with Lys315 in the intramembrane loop and the lining residues on the cytoplasmic pore, providing a platform for the interaction with the BsKtrA octameric ring, known as lateral contact[7]. The C-terminus of BsKtrB remains virtually unchanged in the ADP- and ATP-BsKtrAB structures,

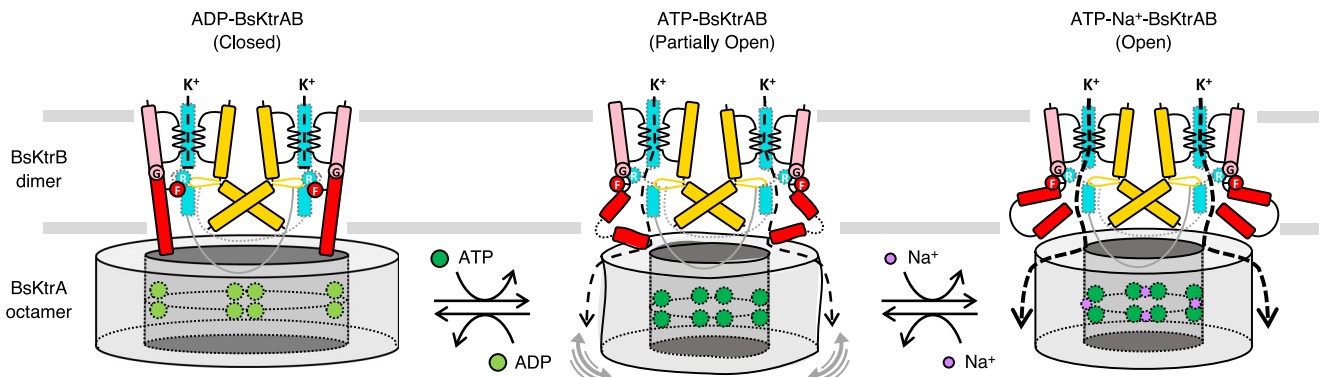

**Fig. 6 | Gating mechanism for BsKtrAB.** Left, ADP binding to BsKtrA induces a diamond-like octameric ring offers the space for BsKtrB D1M2 helix in an extended conformation, resulting in a hydrogen bond interaction of Arg417 (denoted as R) in the middle of D4M2 helix (cyan) with the intramembrane loop, which creates a positively charged barrier for K⁺ passage. In this conformation, the hydrophobic side chain of Phe91 (denoted as F) faces to the permeation pathway, further blocking the aqueous flow in pore. Center, binding of ATP alone to BsKtrA causes instability to the octameric ring and difficulty in proper BsKtrAB assembly. The pore is partially open due to the transient interactions between BsKtrA and BsKtrB.

Right, upon binding of both ATP and Na⁺ to BsKtrA, the square-shaped octameric ring is thermodynamically stabilized, causing a steric hindrance for the D1M2 helix, and resulting in a helical hairpin (red). In this conformation, the side chain of Arg417 alternatively interacts with the backbone oxygen of Gly87 (denoted as G) in the discontinuous region of D1M2 helix, and the side chain of Phe91 swings away from the pore due to the conformational change of helical hairpin, by which the K⁺ transport activity of BsKtrAB is activated. Figure adapted from Fig. 7 in Diskowski et al.[16].

further fastening the conformation of the intramembrane loop. As a consequence, the open channel conformation of ATP-BsKtrAB is achieved mainly by the release of the constraints between D1M2b and the intramembrane loop.

As commented by Inga Hänelt and colleagues[2], the high intracellular ATP/ADP ratio in normal cellular condition and the stronger ATP to KtrA binding affinity implicated that it is difficult for ADP to exert its inhibitory effect. Therefore, additional effectors might be involved in the regulatory process of ATP. It has been shown that in an euryhaline cyanobacterium *Synechocystis*, after being transferred into a saline medium, the intracellular [Na⁺] rapidly increased within a few minutes in the early phase of osmoadaptation, and was subsequently replaced by K⁺ ions, which lasted for hours to days[39]. As Na⁺ ions have a deleterious effect on the function of bacterial cells, a surge of intracellular Na⁺ concentration in hyperosmotic condition needs to be counterbalanced by enhancing the influx of K⁺. It was therefore postulated that Na⁺ is involved in stimulating the uptake of K⁺. In this study, we demonstrated that Na⁺ plays a critical role in the ATP-dependent activation of the BsKtrAB system for K⁺ uptake by enhancing the stability of ATP-BsKtrA octameric ring and strengthening the BsKtrAB complex conformation at an open state. The thermostability assays and functional characterization indicated that binding ATP alone to BsKtrA is actually detrimental for maintaining an open pore state of BsKtrB. In contrast, ADP-BsKtrA is comparably more stable than ATP-BsKtrA if Na⁺ is not bound. The millimolar range of Na⁺ binding affinity for ATP-BsKtrA is consistent with the previously reported concentrations of Na⁺ for KtrAB activation[12,13,40]. The relatively low affinity for Na⁺ implies that the counteracting K⁺ uptake is induced only when the intracellular [Na⁺] is high.

Mg²⁺ and Ca²⁺ have been implicated in the Na⁺-binding site at the intradimer interface of ATP-BsKtrA[7,11]. Mg²⁺ is a key cofactor in ATP hydrolysis. However, to the best of our knowledge, there is no previous study indicating that KtrA possesses the ATP hydrolysis activity, and our DSF studies clearly indicate that Mg²⁺ cannot induce significant change in the thermostability of ATP-BsKtrA. Ca²⁺ elevates the channel activity of MthK through directly binding to its RCK domain, as demonstrated by structural and electrophysiological analyses[41,42]. In this study, Ca²⁺ seems to interact with BsKtrA, using a similar mechanism of Na⁺, to activate the channel activity of BsKtrB. However, the signaling pathway of Ca²⁺ in bacterial cells remains to be

elucidated[43], and the physiological significance of Ca²⁺ in KtrAB system requires a deeper investigation.

The Na⁺-dependent K⁺ uptake systems are also present in animals[21,44]. $K_{Na}$ channels (Slo2.1 and Slo2.2) have received particular attention for their physiological importance in maintaining precise timing of action potential[45] and cell volume[46]. Sequence analysis and functional studies indicated that a consensus motif of NAD⁺-binding site similar to the Rossmann fold of BsKtrA is localized in the RCK2 domain of $K_{Na}$[29], and interestingly, $K_{Na}$ channels were activated by NAD⁺ in a [Na⁺]-dependent manner in a millimolar range[29]. In contrast, ATP seemed to inhibit the activity of Slo2.1[21], but a recent study suggested that ATP has no effect on the activity of Slo2.1[47]. Further characterization is therefore necessary to define the regulatory role of ATP in $K_{Na}$ channels in the presence of Na⁺.

In summary, the high-resolution cryo-EM structures of BsKtrAB at the activated and inactivated states shed light on a mechanism of ligand-gated K⁺ channels. The structural and functional studies illustrating the binding site of Na⁺ of KtrA and its synergistic role in the ATP activation of KtrAB also provide an underlying framework to elucidate the mechanistic model of Na⁺-activated $K_{Na}$ channels in central nervous system neurons.

## Methods
### Protein expression and purification
The procedure for the purification of BsKtrA and BsKtrB was modified based on Vieira-Pires et al. [7]. Tag-less BsKtrA was overexpressed in *E. coli* BL21(DE3) in LB medium containing 100 μg/ml ampicillin at 20 °C for 16 h after 400 μM IPTG induction. Cell pellets were resuspended in Buffer A (50 mM Tris-HCl pH 8.0, 50 mM KCl, 5 mM DTT), and cell lysis was performed using a high-pressure cell disruptor (Constant Systems) at a pressure of 25 kpsi with Buffer A supplemented with Protease Inhibitor Cocktail (Roche), 1 μg/ml DNase (Sigma-Aldrich), and 1 mM MgCl₂. The supernatant of the centrifuged lysate was loaded into an anion exchange column packed with Macro-Prep High Q Resin (Bio-Rad) and washed with Buffer B (50 mM Tris-HCl pH 8.0, 100 mM KCl, 5 mM DTT). The proteins in the lysate were fractionized using a KCl concentration gradient (from 100 mM to 600 mM) in Buffer B. The fractions containing BsKtrA were slowly loaded onto the N⁶-hexyl-ATP agarose (Jena Bioscience) column, which was then washed with Buffer C (50 mM Tris-HCl pH 8.0, 150 mM KCl, 5 mM DTT). BsKtrA was eluted

with Buffer C containing 5 mM ATP or 5 mM ADP followed by a thorough dialysis (four times of 100-fold dialysis) in Buffer D (50 mM Tris-HCl pH 8.0, 150 mM KCl, 1 mM DTT, 1 mM EDTA) to remove bound ATP or ADP and divalent cations.

N-terminal His-tagged BsKtrB with the tobacco etch virus (TEV) protease cleavage site was overexpressed in *E. coli* C43 (DE3) in TB medium containing 100 μg/ml ampicillin at 25 °C for 16 h after 400 μM of IPTG induction. Cell pellets were resuspended and lysed in Buffer E (50 mM Tris-HCl pH 8.0, 120 mM NaCl, 30 mM KCl) supplemented with the protease inhibitor cocktail, 1 μg/ml DNase and 1 mM $MgCl_2$. BsKtrB was extracted with 1 % DDM (n-dodecyl-β-D-maltoside, Carbosynth) in Buffer E at 4 °C for 2 h followed by ultracentrifugation at 150,000 g for 1 h. The supernatant was loaded onto a Ni-NTA (GE Healthcare) affinity chromatography column and washed with Buffer F (50 mM Tris-HCl pH 8.0, 120 mM NaCl, 30 mM KCl, 0.03% DDM). The protein was eluted with Buffer F supplemented with 250 mM imidazole. The fractions containing BsKtrB were dialyzed overnight at 4 °C in the presence of His-tagged TEV protease in Buffer F. The cleaved sample was loaded onto a Ni-NTA His-Trap column (GE Healthcare), and the flow-through containing BsKtrB was collected for further purification.

The purified BsKtrA and BsKtrB were subjected to size exclusion chromatography (SEC) using a Superdex 200 increase 10/300 (GE Healthcare) column for the final polish and buffer exchange. The SEC buffers for BsKtrA and BsKtrB are varied depending on the following experiments (see below). All purified protein samples were snap-frozen in liquid nitrogen and stored at −80 °C.

### Cryo-EM sample preparation
For single-particle cryo-EM structure determination of BsKtrAB complexes, BsKtrA purified with K⁺ Buffer (50 mM Tris-HCl pH 8.0, 150 mM KCl, 1 mM TCEP) and BsKtrB purified in Buffer F were mixed at the BsKtrA to BsKtrB molar ratio (2:1) with the addition of 1 mM ATP or ADP, respectively, and incubated for 2 h. The properly assembled ATP- or ADP-BsKtrAB complex samples were fractionized as monitored by SEC elution profiles using Superdex 200 increase 10/300 column with Cryo-EM Buffer (20 mM Tris-HCl pH 8.0, 70 mM NaCl, 30 mM KCl, 0.75 mM 6-cyclohexyl-1-hexyl-β-D-maltoside).

### Cryo-EM grid preparation and data acquisition
For $Mg^{2+}$-added ATP-BsKtrAB (Structure I), 0.15 mg/ml BsKtrAB was mixed in the Cryo-EM Buffer containing 2 mM $MgCl_2$ and 1 mM ATP. For $Mg^{2+}$-free ATP-BsKtrAB in the presence of EDTA/EGTA (Structure II), 4 mg/ml BsKtrAB was prepared in the Cryo-EM Buffer containing 1 mM ATP, 2 mM EDTA, and 1 mM EGTA. For ADP-BsKtrAB (Structure III), 0.24 mg/ml BsKtrAB was diluted in the Cryo-EM Buffer containing 100 μM ADP. All grids were prepared using the Vitrobot Mark IV (Thermo Scientific) at 4 °C and 100% humidity. 4 μl of protein samples were applied onto the freshly glow-discharged holey carbon films. Quantifoil R1.2/1.3 plus C2, graphene oxide coated UltrAuFoil R2/2, and Quantifoil R2/1 plus C2 were used for $Mg^{2+}$-added ATP-BsKtrAB, $Mg^{2+}$-free ATP-BsKtrAB, and ADP-BsKtrAB, respectively, with a 10 s wait time. The grids were then frozen in nitrogen-cooled liquid ethane after blotting for 3.5 s. All data were acquired on a 300 kV Titan Krios (Thermo Fisher) cryo-transmission electron microscopy equipped with a K3 Summit direct electron detector (Gatan) and GIF Quantum energy filter (Gatan) in super-resolution mode. The detailed parameters are summarized in Supplementary Table 1.

### Cryo-EM image processing
All raw movie stacks were motion-corrected and dose-weighted using MotionCor2[48], which involved a two-fold binning factor, resulting in a pixel size of 0.83 Å per pixel. The motion-corrected movies were then transferred to CryoSPARC for the subsequent processes[49], including CTF estimation, particle picking, 2D classification, 3D classification, and 3D map refinement as shown in Supplementary Figs. 3–5. In brief,

the micrographs with high resolution were selected for particle picking after CTF estimation. Initial particle picking employed Blob Picker were extracted with a 384-pixel box size for subsequent 2D classification. The classes with clear 2D views were adopted as the templates for the second round of particle picking. After three rounds of 2D classification, the remaining good particle stacks were selected for the 3D map ab-Initio reconstruction and classification.

For the $Mg^{2+}$-added ATP-BsKtrAB (Structure I) dataset, 805,032 particles from 2D classification were subjected to ab-initio reconstruction and heterogeneous classification with C1 symmetry. 549,841 particles with $KtrB_2A_8B_2$ assembly from a single class were selected for homogeneous, non-uniform, and local refinement (using an entire protein complex mask) with C2 symmetry, resulting in a 2.48 Å $Mg^{2+}$-added ATP-BsKtrAB map.

For the $Mg^{2+}$-free ATP-BsKtrAB (Structure II) dataset, 755,437 particles from 2D classification were selected for two rounds of ab-initio reconstruction and heterogeneous classification with C1 symmetry. 527,427 particles with $KtrB_2A_8B_2$ assembly were then subjected to homogeneous and non-uniform refinement with C2 symmetry, resulting in a 2.82 Å $Mg^{2+}$-free ATP-BsKtrAB map.

For the ADP-BsKtrAB (Structure III) dataset, 666,155 particles from 2D classification were used for ab-initio reconstruction and heterogeneous classification with C1 symmetry. 340,666 particles with $KtrB_2A_8B_2$ assembly were then subjected to the second round of 3D map generation and classification. 294,844 particles were selected for further homogeneous, non-uniform, and local refinement (using the entire protein mask) with C2 symmetry, resulting in a 2.83 Å ADP-BsKtrAB map. To improve the map quality, focused refinement with masking BsKtrA octameric ring region and BsKtrB dimer region were performed. For BsKtrB dimer, the symmetry was expanded and one KtrB dimer was masked during the refinement process.

### Cryo-EM model building and refinement
For $Mg^{2+}$-added ATP-BsKtrAB, the published crystal structure of ATP-BsKtrAB[7] (PDB code: 4J7C) was used as an initial model and docked into the map using UCSF chimera. All residues from the fitted model were manually checked and refined using COOT[50]. DeepEMhancer sharpening map was applied to assist the model building of RCK_C domain[51]. The model building was then accomplished by several rounds of refinement by *phenix.real_sapce_refine* of the Phenix suite[52] and iteratively building using COOT. The final model of $Mg^{2+}$-added ATP-BsKtrAB was further used as the initial model for the model building of $Mg^{2+}$-free ATP-BsKtrAB. For ADP-BsKtrAB, the BsKtrA model from the published 5.97 Å ADP-BsKtrA$_{ΔC}$B crystal structure[15] (PDB code: 5BUT) and the BsKtrB model from the published 3.50 Å ATP-BsKtrAB crystal structure[7] (PDB code: 4J7C) were used as the initial models and docked into the map by using *phenix.dock_in_map*. For the regions with low-resolution density maps, we employed focused-refined maps of the BsKtrA octamer and BsKtrB dimer to assist the model building process. The model building and refinement procedures were performed as described above. The statistics of the structure refinement were summarized in Supplementary Table 1.

### Full atomistic modeling
The full atomistic structures of ATP- and ADP-BsKtrAB complexes were constructed based on the cryo-EM resolved structures (Structures II and III). The missing residues Gly103 and Lys104 in the ATP-BsKtrAB were generated by homology modeling using SWISS-MODEL[53]. To obtain an initial structure of the missing part of ADP-BsKtrAB (Gly103 to Gly124), CCbuilder[54] was used to generate the D1M2b helix by using the amino acid sequences from Ile98 to Leu118. The initial model of the D1M2b helix in the ADP-BsKtrAB model was obtained by aligning the helix with cryo-EM resolved coordinates of the Ile98, Val99 and Met100. Homology modeling using SWISS-MODEL[53] was then applied to generate the molecular structures of other missing residues in ADP-

BsKtrAB. After obtaining the full atomistic model of the ATP- and ADP-BsKtrAB complexes, we used the CHARMM-GUI server[55] to generate lipid bilayers and construct the protein/membrane complex. TIP3P water was used for all the simulations. The initial model box size is about 200x200x200 Å³ and the KCl concentration is 0.15 mol/L. All the Molecular dynamics (MD) simulations are performed by Nanoscale molecular dynamics (NAMD)[56] with CHARMM force field. Visual molecular dynamics (VMD)[57] is used for the visualization and analysis of the simulation results.

To refine the molecular structures of the ATP- and ADP-BsKtrAB complexes, an energy minimization was performed firstly for the protein/membrane complex. After the energy minimization, an NPT ensemble simulation with a timestep of 1.0 fs is performed for 1 ns with unrelaxed lipid bilayers was implemented followed by 1 ns simulation with relaxed lipid bilayers. Then, the timestep was switched to 2.0 fs for additional 2 ns NPT ensemble simulation to ensure the protein/membrane system is stable. Finally, 45 ns NPT ensemble simulations with 2.0 fs timestep for both ATP- and ADP-BsKtrAB complexes were performed to investigate the structural stability (Supplementary Fig. 21) and differences.

Furthermore, steered molecular dynamics (SMD) simulations were performed by using final equilibrium structures of KtrB from MD simulations to study the molecular gating mechanism. The initial position of the POT is set at 10 Å above the selectivity filter. A spring was attached to the POT and the POT is pulled toward the selectivity filter. The harmonic spring was set as a constant of 7 kcal/mol/Å², and the constant velocity was set at 0.01 Å/ps. In SMD simulations, we restricted the $C_\alpha$ of the endpoint of the helices in the KtrB (Supplementary Fig. 22). The total SMD simulation time is 4 ns with a timestep of 2.0 fs. All of the results were calculated by three independent simulations for validation. For hydrogen bond analysis, the distance cutoff is 3.5 Angstrom and the Angle cutoff is 30 degrees. Three independent simulations were performed for the ATP- and ADP-bound models to calculate the hydrogen bond occupancy.

### Protein crystallization
To obtain the Tl⁺-derivatized ATP-BsKtrA crystals, BsKtrA purified in K⁺ Buffer was concentrated to 10 mg/ml and supplemented with 1 mM ATP dipotassium salt hydrate (KATP) and 10 mM TlOAc. The crystallization condition for the ATP-BsKtrB octamer was modified as previously described[7]. Crystals were grown at 20 °C using hanging-drop vapor diffusion by mixing 1 μl of protein with 1 μl of precipitant. For the sodium competition assay, BsKtrA was purified in Na⁺ Buffer (50 mM Tris-HCl pH 8.0, 150 mM NaCl, 1 mM TCEP), followed by addition of 1 mM ATP disodium salt hydrate (NaATP) and 5 mM TlOAc before crystallization.

### Data collection and structure determination
The X-ray diffraction data were collected using TPS 05A beamline at the National Synchrotron Radiation Research Center (NSRRC) in Taiwan. The diffraction data sets were indexed, integrated and scaled using the HKL-2000 package[58]. Molecular replacement was utilized to solve the structure by Phaser[59] with the published octameric ATP-BsKtrA coordinate[7] (PDB code: 4J90) as the search model. The structure was manually refined using COOT, and further structure refinement was performed with Phenix software suite[52]. Anomalous difference maps of Tl⁺ were calculated using Phenix. The X-ray crystallographic data collection and refinement statistics are summarized in Supplementary Table 2.

### Isothermal titration calorimetry
ITC experiments were performed by using ITC200 calorimeter (MicroCal Inc.) at 25 °C with a 600 rpm stir speed. BsKtrA prepared in Na⁺ Buffer or K⁺ Buffer with a final concentration of 30 μM in a volume of 280 μl was titrated with 300 μM NaATP or 800 μM KATP,

respectively. ITC measurements involved 20 injections of titrants with 2 μl for each injection. Each titration point was subtracted by the control experiments with ATP titration into either of the buffers. The experiments were performed in triplicates and all the titration points except for the first one were analyzed using the MicroCal ITC-Origin.

### LC-MS/MS analysis
To obtain apo-BsKtrA, protein samples were dialyzed in K⁺ Buffer in different folds of dialysis (100-, 10,000- and 1,000,000-fold) followed by further purification using Superdex 200 increase 10/300 (GE Healthcare) in K⁺ Buffer. LC-MS/MS analysis was performed to examine the residual ATP bound to BsKtrA (Supplementary Fig. 23). 10 μg of BsKtrA from each preparation of dialysis folds was denatured at 95 °C for 5 min and the aggregated protein was removed by ultra-centrifugation at $18,000 \times g$ for 30 min. The supernatant was then collected, vacuum-dried, and resuspended in 15 μl of deionized water for triplicate LC-MS/MS analysis. The analysis was performed using an ACQUITY H-Class UPLC system coupled with an XEVO TQ-MS with an ESI ionization source (Waters, Milford). The 1 μl injection of samples were separated through a ZIC-cHILIC column (3 μm, 2.1 mm × 100 mm) at a flow rate of 0.25 ml/min at 35 °C. A binary gradient system consisting of mobile phases A and B, which were 0.1% formic acid (FA) in DI water and 0.1% FA in acetonitrile (J.T. Baker), respectively. The following gradient program was used: 0–2 min, 70% B; 2–2.5 min, 70%–5% B; 2.5–5 min, 5% B; 5–5.1 min, 5%–70% B; and 5.1–10 min, 70% B. The mass spectrometer was operated in negative ionization mode using multiple reaction monitoring (MRM) mode. The MRM transitions monitored were m/z 505 to m/z 158 for ATP, with the following parameters: cone voltage of 28 V, collision energy (CE) of 26 V, and a dwell time of 0.025 s. The following MS parameters were used: capillary voltage of 3.0 kV; desolvation temperature of 350 °C; desolvation gas flow of 600 L/h; collision gas flow of 0.25 ml/min. Both Q1 and Q3 quadrupoles were maintained at quantitative resolution. Peak data visualization and presentation were performed using MassLynx V4.1 software and GraphPad Prism.

### ICP-MS analysis
ICP-MS was utilized to analyze the abundance of metal ions bound to ATP-BsKtrA. 5 mg/ml of either apo-BsKtrA or ATP-BsKtrA in Na⁺ Buffer, K⁺ Buffer, K⁺ Buffer plus 10 mM CaCl₂, or K⁺ Buffer plus 10 mM MgCl₂ were prepared and incubated on ice for 1 h. Subsequently, buffer exchange in Choline Buffer (50 mM Tris-HCl pH 8.0, 150 mM choline chloride, 1 mM TCEP) was performed using a Superdex 200 increase 10/300 column (GE Healthcare). The collected protein fractions (300 μl) were treated with 5 ml of nitric acid. The digestion process was initiated using a microwave accelerated reaction system (CEM MARS 230/60) at 1600 W, with the temperature gradually increasing to 160 °C over 20 min, and held at that temperature for 40 min. The denatured protein samples were then diluted with deionized water to a final volume of 20 ml. The treated protein samples were analyzed with an ICP-MS system (NexION 300X, PerkinElmer), in the DRC mode (collision mode) at the Health Technology Center of Chung Shan Medical University.

### Urea unfolding assay
The urea unfolding assay was performed using the intrinsic tryptophan fluorescence of BsKtrA as described previously[34]. Purified BsKtrA samples in K⁺ Buffer or Na⁺ Buffer were diluted to 1 μM using respective buffers, and titrated with increasing concentrations of K⁺ or Na⁺, respectively, in the presence of 0.1 mM ATP or ADP, according to the experimental design, and then incubated for 1 h on ice. Protein samples were gently mixed with urea to reach different final urea concentrations and incubated at 25 °C for 30 min. The sample mixtures were transferred to a quartz cuvette (Starna scientific, Type 16.10/Q/10) for full-wavelength scanning of the fluorescence derived

from the intrinsic tryptophan, using a spectrofluorophotometer (RF-6000, SHIMADZU) (Excitation: 295 nm, Emission: 310–380 nm). The tryptophan fluorescence emission intensities (330 nm) were normalized against the intensity of protein samples without urea treatment. The normalized fractions representing the percentages of folded BsKtrA were plotted against the concentration of urea. The unfolding concentration ($C_m$) of urea was analyzed using Sigmoidal 4PL model by GraphPad Prism.

### Thermal stability assays
Thermal shifts of BsKtrA were monitored using differential scanning fluorimetry (DSF), as previously described[60]. Briefly, BsKtrA purified in Choline Buffer (50 mM Tris-HCl pH 8.0, 150 mM choline chloride, 1 mM TCEP) was diluted to 5 μM in the presence of 100 μM KATP or KADP with the titration of increasing concentrations of $K^+$, $Na^+$, $Ca^{2+}$ or $Mg^{2+}$ according to the experimental design. Sypro Orange 5000X (Sigma) was added to the mixtures to reach a final concentration of 5X, and the mixtures were placed in a 96-well PCR white plate (Bio-Rad). The assay was performed using a real-time PCR detection system (CFX Connect, Bio-Rad) with FRET channel scan mode. The temperature scan was performed from 25 °C to 85 °C with an increment of 0.3 °C per step with 12 s dwell time. The transition temperature ($T_m$) was analyzed using CFX Manager (Bio-Rad), and then normalized with respect to the minimal (0%) and maximal (100%) change of $T_m$ detected. The normalized fractions were plotted against the concentrations of titrated cations, and the apparent dissociation constant ($K_{dapp}$) was determined by one-site specific binding model using GraphPad Prism.

For protein unfolding half-life determination, the samples prepared as mentioned above were subjected to a real-time PCR thermocycler with a constant temperature at 40 °C. The fluorescence intensity was continuously measured for 175 min. The analysis of protein unfolding half-life was carried out as previously described[61]. The fraction of folded protein was calculated by the equation: $1-F_i/F_{max}$, where $F_i$ is the fluorescence of each time point, and $F_{max}$ is the maximal fluorescence measured in that specific sample. The half-life constants were determined by GraphPad Prism using exponential one-phase decay model.

### BsKtrAB complex assembly assay
Protein complex assembly was monitored using SEC. The elution volumes for BsKtrA octamer, BsKtrB dimer and BsKtrAB complex were benchmarked as demonstrated by Morais-Cabral and coworkers[7]. Individually purified BsKtrA (0.125 mg/ml) and BsKtrB (0.125 mg/ml) were mixed in a molar ratio of 2:1 in either $K^+$ Buffer or $Na^+$ Buffer, supplemented with 0.03% DDM in the presence of 50 μM KATP or NaATP, respectively, for 1 h incubation at 4 °C. The mixtures were subjected to Superdex 200 increase 10/300 column for SEC profile analysis.

### Preparation of proteoliposomes
The preparation was modified based on the previous studies[11]. Briefly, *E. coli* polar lipids (Avanti) were dissolved in ether, followed by evaporation using an argon stream to remove any organic solvent. The lipids were then resuspended in Swelling Low $K^+$ Buffer (10 mM HEPES, 7 mM NMG pH 8.0, 0.2 mM EDTA, 150 mM KCl) to a final concentration of 10 mg/ml, followed by 3 rounds of freeze-thaw cycles in liquid nitrogen. Large unilamellar vesicles (LUVs) were prepared using an extruder (Avanti) with a membrane filter of 400 nm pore size. For proteoliposome reconstitution, the LUV solution was supplemented with 30 mM DM (n-Decyl-b-D-Maltoside, Anatrace), and properly assembled BsKtrAB complex was added to the LUV solution with a protein-to-lipid ratio of 1:100 (w:w) in the presence of 0.1 mM KATP or KADP for a gentle mixing at 25 °C for 1 h. The protein-lipid mixture was incubated twice with fresh SM-2 Biobeads (BioRad) at a bead-to-

detergent ratio of 20:1 (w:w) at 25 °C for 1 h and then at 4 °C for overnight to remove residual detergents.

For the sodium-dependent $K^+$ flux assays, the salt concentration in Swelling Low $K^+$ Buffer for LUV preparation was increased to either 200 mM KCl (Swelling $K^+$ Buffer) or 150 mM KCl and 50 mM NaCl (Swelling $Na^+$ Buffer). Purified BsKtrB was reconstituted into the LUVs with a protein-to-lipid of 1:50 (w:w) as mentioned above, followed by incubation with BsKtrA with a molar ratio of 1:2 (KtrB:KtrA) at 4 °C for 1 h in the presence of 0.1 mM KATP or KADP. The proteoliposomes were then immediately used for $K^+$ flux assays.

### Fluorescence-based $K^+$ flux assay
The fluorescence-based $K^+$ flux assay was carried out on the basis of previously published studies[11]. To establish the $K^+$ gradient, the proteoliposomes were diluted 100-fold in Flux Buffer (10 mM HEPES, 7 mM NMG pH 8.0, 0.2 mM EDTA, 150 mM sorbitol, 0.1 mM KATP or KADP). The samples were transferred to a quartz cuvette with a stirrer for fluorescence measurements at 25 °C using a spectrofluorometer (RF6000, Shimazu). The pH-sensitive dye ACMA (9-amino-6-chloro-2-methoxyacridine, Sigma) was then added to a final concentration of 500 nM and incubated for 5 min. The fluorescence was then monitored every 0.5 s ($\lambda_{ex}$ = 410 nm, $\lambda_{em}$ = 480 nm). The initial baseline was measured for 100 s, and the assay was initiated by adding 2 μM of $H^+$ ionophore CCCP (carbonyl cyanide m-chlorophenyl hydrazine, Sigma) and measured for 400 s. For the final baseline, 300 nM of the $K^+$ ionophore valinomycin (Sigma) was added and an additional 100 s of fluorescence intensity was recorded. For $Na^+$-dependent flux assay, the sorbitol in Flux Buffer was increased to 200 mM, and the dilution of proteoliposomes into the Flux Buffer was increased to 200 folds. The normalization of fluorescence quenching curves and calculation of flux rate constants followed the previously described methods[11]. To normalize the fluorescence quenching curves, each dataset of individual experiment was normalized using the following equation: $NF = (F − F_{val})/(F_{ini} − F_{val})$, where NF is the normalized fluorescence, F is the fluorescence of each time point, $F_{ini}$ is the last baseline point measured before CCCP addition, and $F_{val}$ is the lowest point measured after valinomycin addition The flux rate constants (100–500 s) were determined using the exponential one-phase decay model in GraphPad Prism. The schematic illustration of normalization and analysis are shown in Supplementary Fig. 15.

### Reporting summary
Further information on research design is available in the Nature Portfolio Reporting Summary linked to this article.

## Data availability
The data that support this study are available from the corresponding authors upon request. The cryo-EM maps have been deposited in the Electron Microscopy Data Bank (EMDB) under accession codes EMD-36803 (Structure I); EMD-36804 (Structure II); EMD-38477 (Structure IIa); EMD-38478 (Structure IIb); EMD-36800 (Structure III); EMD-36801 (Structure III, focused refined on KtrA octamer); and EMD-36802 (Structure III, focused refined on KtrB dimer). The atomic coordinates have been deposited in the Protein Data Bank (PDB) under accession codes 8KIT (Structure I); 8K1U (Structure II); 8XMH (Structure IIa); 8XMI (Structure IIb); 8K1S (Structure III); 8K16 ($Tl^+$-treated BsKtrA in $K^+$ Buffer); and 8K1K ($Tl^+$-treated BsKtrA in $Na^+$ Buffer). Source data are provided with this paper.

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

## Acknowledgements

We thank Yuch-Cheng Jean and the staff of beamlines TLS 15A and TPS 05A, National Synchrotron Radiation Research Center (NSRRC) in Taiwan. The work research work was supported by research grants from National Science and Technology Council (NSTC110-2311-B-005 -005 -MY3, NSTC109-2311-B-005-005- and NSTC108-2311-B-005-003- to N.-J.H.). The cryo-EM work was supported by Academia Sinica, Taiwan Protein Project (AS-KPQ-105-TPP and AS-KPQ-109-TPP2 to M.-D.T.), and Taiwan Cryo-EM Consortium funded by National Council of Science and Technology (NSTC 112-2740-B-006-001). The cryo-EM experiments were performed at the Academia Sinica Cryo-EM Center (ASCEM) and the cryo-EM data were processed at the Academia Sinica Grid-computing Center (ASGC). ASCEM is supported by Academia Sinica (AS-CFII-108-110) and Taiwan Protein Project. ASGC is supported by Academia Sinica. We thank Dr. Shu-Chuan Jao in the Biophysics Core Facility, funded by Academia Sinica Core Facility and Innovative Instrument Project (AS-CFII-111-201), for providing technical assistance of ITC experiments. The authors thank Dr. Meng-Chiao Joseph Ho and Wen-Jin Winston Wu for useful discussions.

## Author contributions

W.T.C., C.-Y.L., Y.-C.C., S.-Y.L. and Y.-K.C. conducted molecular cloning, point mutations, protein expression, purification, protein complex assembly, and biophysical studies. W.T.C. designed and performed K⁺ flux assays. W.T.C., C.-Y.L., Y.-C.C. and C.-J.C. performed protein crystallization, X-ray diffraction experiments, anomalous scattering and crystal structure determination. Y.-K.C., C.-H.W., Y.-P.H. and S.P.M. carried out cryo-EM grid preparation, screening, data collection, processing and model building. W.-H.H., S.-W.C., T.-L.H. and R.-S.C. performed MD simulation and computational analysis. W.-C.W. and C.-C.L. carried out the LC MS/MS analysis. S.-H.C. and M.-H.H. assisted experimental designs and data analyses. W.T.C and Y.-K.C. prepared the figures. W.T.C., Y.-K.C., M.-D.T. and N.-J.H. wrote the manuscript. M.-D.T. and N.-J.H. conceived and supervised the research project.

## Competing interests

The authors declare no competing interests.
