## [Peer Review File · Nature Communications]

Structural Basis and Synergism of ATP and Na⁺ Activation in Bacterial K⁺ Uptake System KtrABReviewer #1 (Remarks to the Author):

The study by Chiang and Chang et al. describes a structural and mechanistic basis of Na⁺ regulation of the nucleotide-gated K⁺ channel KtrAB, which forms a quaternary complex of KtrB₂A₈B₂. Using cryo-EM of the KtrAB complex, they identified a density at the ATP-bound KtrA dimer interface, even in the presence of chelating agents; furthermore, using TI⁺ and anomalous scattering of ATP-bound KtrA, they unambiguously identified an TI⁺ at this same site that could be outcompeted by Na⁺ but not K⁺, strongly suggesting that this is an accessible, specific Na⁺ site; previously, this was attributed to that of a divalent cation. Subsequently, the authors use biophysical and functional analyses to identify a stabilizing role of Na⁺ in the ATP-bound KtrAB complex, and further identify a Na⁺-dependence of K⁺ flux. Finally, using the ATP- and ADP-bound structures, the authors propose a gating mechanism where both a positively charged residue and hydrophobic residue restrict the flow of K⁺ ions; ATP binding induces rearrangement of these gates, facilitating K⁺ passage.

It was a pleasure to read this manuscript. For the most part, the data was clearly presented, the text well-written, the logic flowed well, and the experimental approach rigorous and systematic. The proposed mechanism of ion-binding promoting quaternary complex stability and driving ATP-dependent channel activity is intriguing and will be of interest to anyone studying membrane protein structure-function. I do have some easily addressable concerns to hopefully strengthen the manuscript.

Major comments:

- To me, the most attractive part of this paper is clearly connecting the identified Na⁺ binding site in structure to a functional role for Na⁺ in stability and synergism with ATP, and the key to this would be inactivating this functional effect with the structurally relevant E125Q mutation. This is especially important because this is the same site Teixeira-Duarte et al propose for divalent cations. The authors have somewhat validated this in Supplementary Figure 12, but testing E125Q in orthogonal experiments is important to unequivocally establish this - in both Na⁺-dependence of KtrA denaturation (either urea or temperature), and KtrAB assembly (such as gel filtration). In both cases, in presence of Na⁺, E125Q should behave near identically to wild-type ATP-KtrAB in K⁺. These E125Q experiments all merit being part of main text figures. It would also be useful to test this with T_m of Ca²⁺ binding to confirm that the stabilization effect is specifically due to this site.
- The Na⁺ densities in the EM structures appears to be right on the plane of the C₂ symmetry axis (by eye, it appears to be horizontal through KtrA), which is of slight concern. As an additional refinement validation, could the C₂ symmetry axis be applied vertically at the KtrB dimer interface instead? Also, if the resolution is good enough in C₁, the authors should also demonstrate that the density is visible here.
- In the EM density corresponding to 8K1U, there is clearly asymmetry of the proposed Na⁺ density (with chains A/B and C/D of KtrA having less pronounced Na⁺ density at high contour levels, than E/F and G/H). This potentially suggests that the Na⁺ sites are partially occupied, which would be in agreement of the proposed mM affinity of Na⁺. Given 30 mM Na⁺ in the buffer, this is likely subsaturating concentrations. Perhaps further processing could cleanly delineate fully Na⁺ bound vs. unbound vs. asymmetric conformations - have the authors tried to do this? I would suggest trying ab initio at higher starting resolutions, heterogeneous refinement with two good densities, or symmetry expansion at C₂ followed by either 3D variability or focused 3D classification without alignment.
- If this proves to be fruitful, the authors should try to compare chains with and without this pronounced density (such as when aligning A to E, or B to F) - upon cursory inspection, I can see subtle backbone changes which are relevant at such a high resolution. Perhaps such comparisons can give insight to the structural changes that Na⁺ induces to increase the stability of the complex.
- It is unclear how the authors confirmed the stability of the structure after equilibration but before the steered MD simulations. A standard intermediate step would be classic MD simulations to ensure stability of the system and assess the equilibration. If this was performed, please indicate this more clearly in the text and figures.

- The authors clearly show that in detergent, the complex is unstable in the absence of Na⁺. However, as currently presented, it is unclear if such strong Na⁺-dependent complex stabilization occurs in the lipid bilayer; the complex may be perfectly assembled in these conditions regardless of Na⁺. It is possible that Na⁺-dependent K⁺-flux may occur through other pathways in addition to complex assembly, or may subtly rigidify the complex instead of completely promoting assembly. The authors should consider this caveat in their results and discussion sections.

Minor comments:

- Line 231: The authors suggest that a three-fold change in ATP affinity in Na⁺ vs. K⁺ is significant – but binding affinities are logarithmic phenomena with regards to ligand concentration, and thus three-fold changes are relatively subtle and insignificant without many replicates (Hulme and Trevethick, BJP 2010). Also, such a subtle change could easily be induced nonspecifically by different chaotropic effects between the different alkali cations, rather than occupancy of the Na⁺ site. Thus, I find it unlikely that Na⁺ is affecting ATP affinity in any substantial, mechanistic way and this discussion should be toned down.

- I find it notable that in the thermal melt experiments, adding ATP to apo protein destabilizes the protein (unlike ADP), and Na⁺ addition recovers this destabilization. I think this is an important point that should be highlighted much better in the text and as its own figure panel, since it is very consistent with the proposed mechanism.

- Does KtrA remain a dimer when purified in apo conditions, for the stability assays? What do the authors think is being measured – increased dimerization, which in turn stabilizes monomer unfolding? Representative gel-filtration profiles would be helpful to address this point.

- Since the outer parts of the complex map are much lower quality compared to the inner parts, the authors should try some better sharpening techniques such as DeepEMhancer to improve these lower quality regions for model building. After model building, the final refinement should be performed on the unsharpened map. Further classification of heterogeneity, as described earlier in my review, may also help this.

- In all plots with standard deviation calculated, individual replicates should also be shown.

- Lines 253-254: please report Mg²⁺ amount, even if N.D., in the text.

- I think it is important to clearly delineate, in both the text and figures, which experiments are being done in KtrA vs the KtrAB complex. Though the authors made good efforts to do this, as someone not familiar with this system, it still took me some time to fully comprehend this distinction. I recommend showing these experiments into two separate figures and clarifying in subsection header (line 192) that these experiments were done in KtrA.

- For clarity, it would be appropriate to keep only the normalized flux figure in the main text and move the non-normalized figure to a supplement. As part of this supplement, a visual explanation of how the data is normalized (to accompany lines 717-720) would also be helpful to a general reader.

- Figure 3c inset, Supplementary Figure 11c: NanoDSF is an inappropriate experiment to measure K_d, since K_d is an equilibrium constant, and these measurements are not performed at equilibrium (<https://doi.org/10.7554/eLife.57264>). These graphs are fine as qualitative measures, but at best should be addressed as an apparent K_d (K_{dapp}).

- I would like to a figure for the raw data for the Ca²⁺ and Mg²⁺ experiments, similar to Supplementary Figure 8.

- Discussion: the authors should have a deeper conversation about their results in the structural, functional, and physiological context of Teixeira-Duarte et al. How does the proposed coordination compare to Ca²⁺ and Mg²⁺? Are both Ca²⁺ and Na⁺ physiologically relevant, or is one more likely than another? Why is there such a stark discrepancy regarding the effect of Mg²⁺, and generally their results regarding divalent cations?

Reviewer #2 (Remarks to the Author):

This manuscript details the structural and biophysical characterization of the activation mechanism of the K⁺ channel KtrAB from *Bacillus subtilis*. Using a range of approaches, the authors conclude that the density assigned as Mg²⁺ or Ca²⁺ in previous structures is likely a Na⁺ ion, and that the binding of Na⁺ has a synergistic effect with ATP binding, which leads to enhanced activation of K⁺ flux through the protein. The authors perform a thorough analysis of their structures in the ATP and ADP-bound states and present a mechanism by which ATP and Na⁺ binding is coupled and how these events affect channel gating properties.

The manuscript is well written overall, and very nicely presented. The structural data presented largely supports the conclusions. However, the functional data presented raises many questions (see below) and are not as supportive of some of the conclusions.

Specific comments

Lines 61-74.

The authors describe the structure of KtrAB in the context of the previous structural studies, which is fine as it contextualizes the new findings from this work. However, this is a complex protein, and I did find this description difficult to follow without a figure to accompany it. The authors only reference their figure 1a in this section, which is not sufficient for a reader unfamiliar with this protein. To clarify this, I would suggest an additional figure, or make reference to the rest of figure 1, which is colour coded to match the schematic in 1a.

The clarity of this description is also diminished by use of KtrAB-specific jargon, e.g. "D3M2", which is not shown on a figure.

Line 84.

The authors make reference to Slo2.1 and Slo2.2 both here and in the Discussion. While the text indicates a functional link between these proteins and KtrAB (i.e. Na⁺-activated K⁺ flux), the presence of this comparison also suggests that they are both structurally and mechanistically linked. However, to my understanding that the only shared domain between these 2 proteins is the RCK domain. I think it would be useful to clarify the text in the introduction to spell out the similarities between these systems and how understanding KtrAB activation could inform on Slo2.1 and 2.2 mechanism.

I think the first section (p.5-9), in which a variety of structural approaches (and ICP-MS) are used to convincingly argue that the mystery density in the structures is a Na⁺ ion, is clearly described and well laid out.

The authors use a urea-mediated unfolding assay to compare the stabilizing effects of Na⁺ and K⁺. I am a bit confused about the rationale behind the experimental design. In the Na⁺ titration (3a), the protein is in buffer containing 150 mM K⁺, and in the K⁺ titration the protein is in buffer containing 150 mM Na⁺. If the goal of this assay was to see if Na⁺ and K⁺ can stabilise the protein, would it not have been better to have the protein in buffer containing a functionally inert cation, like choline? In the presence of the 150 mM Na⁺, the protein looks to be "fully stabilized" by Na⁺ binding, so how did the authors expect to see any further stabilization by addition of K⁺?

The authors use a SYPRO-based DSF assay to monitor the stabilizing effects of Na⁺ and K⁺, which convincingly shows that there is a dose-dependent stabilization by Na⁺, but not by K⁺. The authors then use the ΔT_m values to plot a binding curve to derive a K_d . While this analysis looks tempting, these data cannot reliably be used to derive affinities because the ΔT_m data are collected at different temperatures, and it cannot be assumed that the binding affinity is constant at different temperatures.

There are ways of doing it (see Bai et al. Scientific Reports 9, article number 2650 (2019)), but this is not the approach used here. The qualitative conclusions drawn do not need this additional analysis anyway.

Line 242.

The authors state they have "solid evidence that Na⁺ plays a critical role in binding of ATP to BsKtrA." I would argue the complete opposite based on the ITC data provided. The authors only observed a 3.2-fold increase in the K_d for ATP in the presence of 200 mM Na⁺. These data clearly demonstrate that KtrA is perfectly capable of binding ATP at a reasonable high affinity in the absence of Na⁺.

Line 249.

See previous comment about deriving K_ds from DSF.

Line 272-279 The K⁺ flux assay.

The data presented by the authors to this point in the manuscript suggests that K⁺ does not interact with KtrAB, so would not be expected to activate the channel, and yet, the authors observe enhanced K⁺ flux in the presence of K⁺ (a substantial 20% enhancement). This needs to be explained and discussed.

From this assay, the authors observe flux activity under all conditions tested, so this is not really ATP/Na⁺-based activation, it is only activity enhancement, and not really much enhancement either – the presence of Na⁺ and ATP has less than a 2-fold stimulation of activity. Do the authors find this surprising? Is this amount of stimulation physiologically significant?

An important control missing from here is the amount of flux in protein-free liposomes (I note that this control is present in figure 4f, but those results also have some marked differences to these data – see below).

Figure 4f. The authors should indicate in the figure legend or figure what the black and white arrows indicate (also needed in Figure 3f).

Please could the authors explain why this activity assay only reaches 0.4 before plateauing, whereas the previous data in Figure 3f reached effectively zero. This behaviour is quite different, but 2 of the conditions are the same.

REVIEWER COMMENTS

Reviewer #1 (Remarks to the Author):

The study by Chiang and Chang et al. describes a structural and mechanistic basis of Na⁺ regulation of the nucleotide-gated K⁺ channel KtrAB, which forms a quaternary complex of KtrB2A8B2. Using cryo-EM of the KtrAB complex, they identified a density at the ATP-bound KtrA dimer interface, even in the presence of chelating agents; furthermore, using Tl⁺ and anomalous scattering of ATP-bound KtrA, they unambiguously identified an Tl⁺ at this same site that could be outcompeted by Na⁺ but not K⁺, strongly suggesting that this is an accessible, specific Na⁺ site; previously, this was attributed to that of a divalent cation. Subsequently, the authors use biophysical and functional analyses to identify a stabilizing role of Na⁺ in the ATP-bound KtrAB complex, and further identify a Na⁺-dependence of K⁺ flux. Finally, using the ATP- and ADP-bound structures, the authors propose a gating mechanism where both a positively charged residue and hydrophobic residue restrict the flow of K⁺ ions; ATP binding induces rearrangement of these gates, facilitating K⁺ passage.

It was a pleasure to read this manuscript. For the most part, the data was clearly presented, the text well-written, the logic flowed well, and the experimental approach rigorous and systematic. The proposed mechanism of ion-binding promoting quaternary complex stability and driving ATP-dependent channel activity is intriguing and will be of interest to anyone studying membrane protein structure-function. I do have some easily addressable concerns to hopefully strengthen the manuscript.

Major comments:

1. To me, the most attractive part of this paper is clearly connecting the identified Na⁺ binding site in structure to a functional role for Na⁺ in stability and synergism with ATP, and the key to this would be inactivating this functional effect with the structurally relevant E125Q mutation. This is especially important because this is the same site Teixeira-Duarte et al propose for divalent cations. The authors have somewhat validated this in Supplementary Figure 12, but testing E125Q in orthogonal experiments is important to unequivocally establish this - in both Na⁺-dependence of KtrA denaturation (either urea or temperature), and KtrAB assembly (such as gel filtration). In both cases, in presence of Na⁺, E125Q should behave near identically to wild-type ATP-KtrAB in K⁺. These E125Q experiments all merit being part of main text figures. It would also be useful to test this with T_m of Ca²⁺ binding to confirm that the stabilization effect is specifically due to this site.

Author response: We thank the reviewer's critical comments. We performed DSF assays using ATP-BsKtrA_{E125Q} in the presence of Na⁺ and K⁺ (Supplementary Fig. 11b and Fig. 3d) or Ca²⁺ (Supplementary Fig. 14g,h). The T_m values of ATP-BsKtrA_{E125Q} remained almost unchanged in Na⁺ and Ca²⁺ titrations, further highlighting the importance of Glu125 of ATP-BsKtrA in the coordination of Na⁺ and Ca²⁺.

We also performed SEC to monitor BsKtrAB complex assembly. The SEC profiles (Fig. 4a) demonstrated that ATP-BsKtrA_{E125Q} completely failed to form a functionally active BsKtrAB complex in the presence of Na⁺ due to lacking of Glu125 side chain for Na⁺ coordination. The above-mentioned results provide a strong piece of evidence that E125 is involved in the binding of specifically Na⁺ and Ca²⁺.

Fig. 4a

2. The Na⁺ densities in the EM structures appears to be right on the plane of the C2 symmetry axis (by eye, it appears to be horizontal through KtrA), which is of slight concern. As an additional refinement validation, could the C2 symmetry axis be applied vertically at the KtrB

dimer interface instead? Also, if the resolution is good enough in C1, the authors should also demonstrate that the density is visible here.

Author response: We appreciate the reviewer's careful examination of the Na⁺ densities which are on the horizontal C2 symmetry axis in EM structures. In response, we re-examined our data and conducted additional refinements with a vertical C2 symmetry axis between the KtrB dimer interface, oriented perpendicularly to the membrane plane (Supplementary Fig. 8b-e). The map refined with the vertical C2 symmetry axis showed comparable Na⁺ densities compared to the original map with a horizontal C2 symmetry axis (Supplementary Fig. 9e-h). We also performed an additional refinement without imposing any symmetry (C1) (Supplementary Fig. 8f-i) and the Na⁺ densities remained visible (Supplementary Fig. 9i-l). We have incorporated these findings into the revised manuscript (line 138-149, Supplementary Fig. 8 and Fig. 9) providing a piece of evidence of the Na⁺ densities in the EM structures. The refinement statistics for the models derived from the additional map reconstructions are summarized in Supplementary Table 1.

Supplementary Fig. 8 Additional map reconstructions of Mg²⁺-free ATP-BsKtrAB with a vertical C2 symmetry axis and without imposing symmetry (C1). **a.** The original map reconstruction of structure II with a horizontal C2 symmetry axis. **b.** The additional map reconstruction of structure II using a vertical C2 symmetry axis resulted in a new model, structure IIa. **c.** Angular distribution of particles used in the final map reconstruction of structure IIa. The

heat map represents the number of particles for different orientations. **d.** Gold standard FSC curve of the 3D reconstruction of structure IIa map. The resolution was demarcated by the criterion of FSC = 0.143. **e.** The cryo-EM density map of structure IIa was refined to 2.85 Å. The local resolution is indicated as the color gradient. **f.** The additional map reconstruction of structure II without imposing symmetry (C1) resulted in a new model, structure IIb. **g.** Angular distribution of particles used in the final map reconstruction of structure IIb. The heat map represents the number of particles for different orientations. **h.** Gold standard FSC curve of the 3D reconstruction of structure IIb map. The resolution was demarcated by the criterion of FSC = 0.143. **i.** The cryo-EM density map of structure IIb was refined to 3.00 Å. The local resolution is indicated as the color gradient.

Supplementary Fig. 9 The Na⁺ binding sites in the ATP-BsKtrAB cryo-EM structures based on different symmetry map reconstructions. Upper panel, a close-up view of the intra-dimer interfaces of ATP-BsKtrA from ATP-BsKtrAB cryo-EM structures, Structure II based on the horizontal C2 map (**a-d**), Structure IIa based on the vertical C2 map (**e-h**) and Structure IIb based on the C1 map (**i-l**). The cryo-EM density maps (gray mesh) are contoured at 10.0 σ in (**a-h**) and 6.0 σ in (**i-l**). The coordinating amino acid side chains and ATP γ -phosphates are shown in stick from each protomer of BsKtrA dimer. Lower panel, the coordination geometries of the assigned Na⁺ cations after structure refinement are depicted as dashed lines with coordination distances indicated.

3. In the EM density corresponding to 8K1U, there is clearly asymmetry of the proposed Na⁺ density (with chains A/B and C/D of KtrA having less pronounced Na⁺ density at high contour levels, than E/F and G/H). This potentially suggests that the Na⁺ sites are partially occupied, which would be in agreement of the proposed mM affinity of Na⁺. Given 30 mM Na⁺ in the buffer, this is likely subsaturating concentrations. Perhaps further processing could cleanly delineate fully Na⁺ bound vs. unbound vs. asymmetric conformations - have the authors tried to do this? I would suggest trying ab initio at higher starting resolutions, heterogeneous refinement with two good densities, or symmetry expansion at C2 followed by either 3D variability or focused 3D classification without alignment.

Author response: We appreciate the detailed analysis by the reviewer of the EM density corresponding to 8K1U (structure II) and the insightful comments on the observed asymmetry in the proposed Na⁺ density. In response to your recommendations, we diligently performed additional data processing and analysis.

Initially, we attempted to try heterogeneous refinement; however, we are unable to obtain the fully Na⁺ bound and fully unbound classes. Subsequently, we proceeded with 3D variability and 3D classification, masking either the overall complex, KtrA octamer, or RCK_N domain of the KtrA octamer. Despite these efforts, the four Na⁺ densities from the KtrA octamer remained asymmetric.

To further investigate, we focused on the RCK_N domain of a single KtrA dimer from the KtrA octamer and conducted 3D classification, as shown below. In the 10 divided classes, class 10 (10.3% particles) demonstrated the weakest Na⁺ density, showing no observable density at the contour level of 5 σ , but a weak density can be observed at a contour level of 4 σ .

4. If this proves to be fruitful, the authors should try to compare chains with and without this pronounced density (such as when aligning A to E, or B to F) – upon cursory inspection, I can see subtle backbone changes which are relevant at such a high resolution. Perhaps such comparisons can give insight to the structural changes that Na⁺ induces to increase the stability of the complex.

Author response: As mentioned above, we observed a class (class 10) with a weaker Na⁺ density from the focused 3D classification. We then conducted a comprehensive structural analysis by aligning the structure from class 10 to the structures of other classes. However, the superposition analysis revealed no significant differences in the structures, suggesting the backbone changes are not as pronounced as initially anticipated. The structure of class 10 shows a very similar conformation compared to those from other classes. The r.m.s.d. ranges from 0.119 to 0.148 over 243 C α atoms, as shown below.

We also checked the structures of the KtrA dimer from Structure II (see below). Please note that Structure II has been updated after revision, with a more complete model building in the RCK_C domain based on the DeepEMhancer sharpening map (Supplementary Table 2). This will be discussed in the next section at Minor comment 4. In the map of Structure II, where the horizontal C2 symmetry was imposed during map reconstruction, chains A/B and C/D exhibited a less pronounced Na⁺ density compared to chains E/F and G/H. However, the superimposition of the structures of chains A/B and E/F further confirmed their high degree of similarity, with an r.m.s.d. of 0.086 over 430 C α atoms, as shown below. This is consistent with the observations from the focused 3D classification, indicating no obvious conformational changes.

5. It is unclear how the authors confirmed the stability of the structure after equilibration but before the steered MD simulations. A standard intermediate step would be classic MD simulations to ensure stability of the system and assess the equilibration. If this was performed, please indicate this more clearly in the text and figures.

Author response: Thank you for your comment. Indeed, classic MD simulations are performed to ensure the stability of the system before the steered MD simulations. We have carefully examined the RMSD of the whole protein (Supplementary Fig. 21a), KtrB (Supplementary Fig. 21b) and relevant domain (R417, G87 and T310) (Supplementary Fig. 21c). The results indicated that the system and the protein structure are indeed equilibrated after the 45 ns MD simulations.

6. The authors clearly show that in detergent, the complex is unstable in the absence of Na⁺. However, as currently presented, it is unclear if such strong Na⁺-dependent complex stabilization occurs in the lipid bilayer; the complex may be perfectly assembled in these conditions regardless of Na⁺. It is possible that Na⁺-dependent K⁺-flux may occur through other pathways in addition to complex assembly, or may subtly rigidify the complex instead of completely promoting assembly. The authors should consider this caveat in their results and discussion sections.

Author response: As the reviewer's request, we performed a KtrAB complex formation assay in lipid bilayer. We reconstituted BsKtrB proteoliposomes and incubated with ATP-BsKtrA (KtrA to KtrB molar ratio 1:1) in the presence of (a) Na⁺ and (b) K⁺. After 60 min incubation, proteoliposomes were pelleted down followed by resuspension. 10 ul of resuspended solution (triplicates) were loaded to SDS-PAGE gel for densitometry analysis. ATP-BsKtrA revealed a non-specific interaction with empty liposome in both Na⁺ and K⁺ buffers and we consider this as background. Upon incubation with KtrB proteoliposome, one can notice an increase of BsKtrA associated with BsKtrB proteoliposome in the pellets. The densitometry intensity of KtrA_{background} was subtracted from the KtrA_{totalbound} and normalized against the sum of KtrB_{dimer} and KtrB_{monomer}. KtrA revealed a significant increase of KtrB proteoliposome interaction in the presence of Na⁺.

In this study, we performed an in-depth stability characterization of BsKtrA in ATP/ADP-bound states in the presence of ATP or ADP, but we did not really characterize the stability of BsKtrAB complex in individual ligand-binding state. Instead, we evaluated the complex assembly efficiency by monitoring the SEC profiles (Fig. 4a). In the given protein incubation time, ATP-BsKtrA in the presence of Na⁺ showed the best efficiency to produce monodisperse distribution of complex species representing an “activated” ATP-BsKtrAB complex, as demonstrated by the cryo-EM structure (Structure II) collected at the specific fraction indicated by the black arrow in Fig. 4a, compared to the sample in the absence of Na⁺ or apo-BsKtrA. The results suggested that only when BsKtrA binds to ATP and gets stabilized by Na⁺, it can form a perfect square-shaped octameric ring and efficiently assemble with BsKtrB and form a “open” pore conformation as the cryo-EM structure (Structure II) demonstrated. In the conformation, ATP-BsKtrA and BsKtrB form a stable complex through two physical contacts: tip contact and lateral contact (Vieira-Pires et al. 2013 Nature). Therefore, Na⁺ can facilitate and rigidify the functionally-activated complex assembly in this context. If BsKtrA fails to form a square-shaped octameric ring, due to different ligand-binding states or loss of neutralization in the absence of Na⁺, BsKtrA probably show a stochastic conformation. In the scenario, BsKtrA can still interact with BsKtrB, as shown in the SEC profiles, with unknown assembly modes, but the complex may not be able to trigger an “open conformation” of the BsKtrB. Therefore, we believe Na⁺ not only enhances interaction/assembly in detergent micelles or lipid bilayer, but, more importantly, it facilitates the functionally active assembly of ATP-BsKtrAB. We have revised the manuscript in the results and discussion (Line 286-294 and 424-437).

Minor comments:

7. Line 231: The authors suggest that a three-fold change in ATP affinity in Na⁺ vs. K⁺ is significant – but binding affinities are logarithmic phenomena with regards to ligand concentration, and thus three-fold changes are relatively subtle and insignificant without many replicates (Hulme and Trevethick, BJP 2010). Also, such a subtle change could easily be induced nonspecifically by different chaotropic effects between the different alkali cations, rather than occupancy of the Na⁺ site. Thus, I find it unlikely that Na⁺ is affecting ATP affinity in any substantial, mechanistic way and this discussion should be toned down.

Author response: We totally agree the reviewer’s comment. We believe the order of binding to BsKtrA would be first ATP and then Na⁺, because the nucleotide binding site at the RCK_N

domain secures the binding of ATP first and then two adjacent ATP molecules provide the site for Na⁺ binding. The ITC experiments were performed with BsKtrA in Na⁺ buffer titrated with ATP. Na⁺ would not contribute to a significant enhancement of ATP affinity to BsKtrA and three-fold increase in affinity is indeed marginal. We have revised the statement (line 248-252).

8. I find it notable that in the thermal melt experiments, adding ATP to apo protein destabilizes the protein (unlike ADP), and Na⁺ addition recovers this destabilization. I think this is an important point that should be highlighted much better in the text and as its own figure panel, since it is very consistent with the proposed mechanism.

Author response: We thank the reviewer for the positive comment.

9. Does KtrA remain a dimer when purified in apo conditions, for the stability assays? What do the authors think is being measured – increased dimerization, which in turn stabilizes monomer unfolding? Representative gel-filtration profiles would be helpful to address this point.

Author response: As the reviewer's request, we provide the gel-filtration profiles (Superdex-200 10/300) of ATP-, ADP- and apo-BsKtrA, as shown below. The three protein samples were purified at 4°C and showed a monodisperse distribution representing a complex size of octamer on the basis of retention volume (200 KDa vs 25 KDa for KtrA monomer). It's important to note that the protein half-life stability assay was performed at 40°C and the DSF assay was a thermal shifting (25-85°C) assay, BsKtrA octamer may fall apart and become dimers or even monomers, which would interfere the assembly with BsKtrB.

Na⁺ induced substantial stabilization of ATP-BsKtrA in both half-life stability and DSF assay. This is attributed to the binding of ATP and Na⁺ across two KtrA monomers (Fig.2a, 2c and 2e), rigidifying the dimerization of BsKtrA. Considering BsKtrA octamer is assembled by four KtrA dimers, the increased dimerization potency of BsKtrA may facilitate BsKtrA octamerization. In contrast, in the presence K⁺, ATP-BsKtrA is unstable because the absence of Na⁺ results in the failure of neutralizing the repulsive force between the γ -phosphates of the two ATP molecules, thereby making KtrA octameric ring assembly unstable.

10. Since the outer parts of the complex map are much lower quality compared to the inner parts, the authors should try some better sharpening techniques such as DeepEMhancer to improve these lower quality regions for model building. After model building, the final refinement should be performed on the unsharpened map. Further classification of heterogeneity, as described earlier in my review, may also help this.

Author response: We appreciate the reviewer's guidance. We applied DeepEMhancer to improve the quality of the outer parts of the map. Model building was extended further in the RCK_C domain of KtrA as shown in updated Supplementary Table 2. After map sharpening by DeepEMhancer, the RCK_C domain for some chains in Structures I and II has been successfully reconstructed, while for structure III, the RCK_C domain could not be rebuilt due to its extremely poor density. Consequently, we have updated the PDB entries for Structures I and II. However, it's important to note that the Na⁺ density itself is relatively weak in unsharpened map. After applying DeepEMhancer, the Na⁺ density may be perceived as noise and become less clear. Despite this, DeepEMhancer demonstrated a significant enhancement in regions with initially weak density, allowing for improved model building. We have deposited the DeepEMhancer sharpening maps into EMDB as additional maps.

11. In all plots with standard deviation calculated, individual replicates should also be shown.

Author response: We've added individual replicates in all the plots (Fig. 3c-e, 4c, 5e and 5f; Supplementary Fig. 11e, 11f, 14d, 14e, 14h and 16b).

12. Lines 253-254: please report Mg²⁺ amount, even if N.D., in the text.

Author response: We've added the measured concentration of Mg²⁺ in the manuscript (Line 273).

13. I think it is important to clearly delineate, in both the text and figures, which experiments are being done in KtrA vs the KtrAB complex. Though the authors made good efforts to do this, as someone not familiar with this system, it still took me some time to fully comprehend this distinction. I recommend showing these experiments into two separate figures and clarifying in subsection header (line 192) that these experiments were done in KtrA.

Author response: Thanks for the suggestion. We have split Fig. 3 into two figures for clarity. The stability assays (urea unfolding, DSF and half-life assays) of KtrA are shown in Fig. 3, and the assembly assay and K⁺ flux assays of KtrAB are now shown in Fig. 4. The original figures, Fig. 4 and Fig. 5, now become Fig. 5 and Fig. 6, respectively in the revised manuscript. We have also changed the header as "Synergistic effects of Na⁺ and ATP on BsKtrA" (line 207)

14. For clarity, it would be appropriate to keep only the normalized flux figure in the main text and move the non-normalized figure to a supplement. As part of this supplement, a visual explanation of how the data is normalized (to accompany lines 717-720) would also be helpful to a general reader.

Author response: As the reviewer's request, we have standardized the data normalization in all of the K⁺ flux assays (Fig. 4b) in the manuscript. Please also refer to the revised Method section (Line 787-793) and the schematic representation of data normalization and flux rate analysis (Supplementary Fig. 15)

15. Figure 3c inset, Supplementary Figure 11c: NanoDSF is an inappropriate experiment to measure K_d , since K_d is an equilibrium constant, and these measurements are not performed at equilibrium (<https://doi.org/10.7554/eLife.57264>). These graphs are fine as qualitative measures, but at best should be addressed as an apparent K_d (K_{dapp}).

Author response: We appreciate the suggestion and we have changed the term as K_{dapp} in the main text.

16. I would like to a figure for the raw data for the Ca^{2+} and Mg^{2+} experiments, similar to Supplementary Figure 8.

Author response: We have provided the raw data of Ca^{2+} and Mg^{2+} DSF experiments and added the raw data in Supplementary Fig. 14a-c.

17. Discussion: the authors should have a deeper conversation about their results in the structural, functional, and physiological context of Teixeira-Duarte et al. How does the proposed coordination compare to Ca^{2+} and Mg^{2+} ? Are both Ca^{2+} and Na^{+} physiologically relevant, or

is one more likely than another? Why is there such a stark discrepancy regarding the effect of Mg²⁺, and generally their results regarding divalent cations?

Author response: According Teixeira-Duarte et al. (2019), the authors assigned Mg²⁺ at the site mainly from the crystallographic perspectives: octahedral coordination geometry and coordination distance, although the coordination distances (2.20Å~2.60Å) after refinement are longer than the conventional coordination distances for Mg²⁺ (2.00Å~2.15Å) (Harding, Acta Crystallogr D Biol Crystallogr 1999). Although not specifically mentioned in Teixeira-Duarte et al., it is believed that they assigned Mg²⁺ at the site because Mg²⁺ is an important cofactor for nucleotide catalyzing enzymes. However, to our best knowledge, no previous studies have demonstrated that BsKtrA can hydrolyze ATP. From our crystallographic and cryo-EM structures, ATP remains intact in complex with BsKtrA, further indicating that BsKtrA cannot catalyze ATP hydrolysis. Therefore, we believe Mg²⁺ is unlikely to occupy this site. However, it cannot eliminate the possibility Mg²⁺ can bind directly to BsKtrB and exert its regulatory effect according to the fluorescence-based K⁺ flux assays (Fig. 6) in Teixeira-Duarte et al, suggesting Mg²⁺ can enhance the K⁺ flux activity of BsKtrB in the absence of BsKtrA or in complex with BsKtrA E125Q.

In the K⁺ flux assays (Fig. 7) of Teixeira-Duarte et al, they demonstrated that Ca²⁺ can enhance the activity of BsKtrAB in an ATP-dependent manner. The results are in agreement with our findings. They also demonstrated the crystal structure of Ca²⁺-bound ATP-BsKtrA with the coordination distances (2.37Å~2.53Å) after refinement, which are within the range of conventional coordination distances for Ca²⁺.

Ca²⁺ has been known to activate a bacterial K⁺ channel MthK through its RCK domain (Smith et al. Nat Commun 2013). Ca²⁺ directly activates MthK in an ATP/ADP-independent manner with [Ca²⁺] for activation of 1 mM. The physiological significance of Ca²⁺ in the activation mechanism of MthK was rarely addressed or studied at cellular level because it remains unclear about the regulatory role Ca²⁺ in bacterial cells.

In contrast, a number of early studies have demonstrated that K⁺ uptake activity of KtrAB is Na⁺-dependent. Some early research studies even believed that KtrAB is a Na⁺-dependent secondary active transporter. Based on our structural and biophysical experiments, along with the in vivo physiological studies, we strongly believe the site in BsKtrA is specific for Na⁺ binding. Although the site is also geometrically favorable for Ca²⁺, it is unfavorable for K⁺ and Mg²⁺. We have revised the discussion with regard to the possible binding of Mg²⁺ and Ca²⁺ to BsKtrA (Line 484-492).

Reviewer #2 (Remarks to the Author):

This manuscript details the structural and biophysical characterization of the activation mechanism of the K⁺ channel KtrAB from *Bacillus subtilis*. Using a range of approaches, the authors conclude that the density assigned as Mg²⁺ or Ca²⁺ in previous structures is likely a Na⁺ ion, and that the binding of Na⁺ has a synergistic effect with ATP binding, which leads to enhanced activation of K⁺ flux through the protein. The authors perform a thorough analysis of their structures in the ATP and ADP-bound states and present a mechanism by which ATP and Na⁺ binding is coupled and how these events affect channel gating properties.

The manuscript is well written overall, and very nicely presented. The structural data presented largely supports the conclusions. However, the functional data presented raises many questions

(see below) and are not as supportive of some of the conclusions.

Specific comments

1. Lines 61-74.

The authors describe the structure of KtrAB in the context of the previous structural studies, which is fine as it contextualizes the new findings from this work. However, this is a complex protein, and I did find this description difficult to follow without a figure to accompany it. The authors only reference their figure 1a in this section, which is not sufficient for a reader unfamiliar with this protein. To clarify this, I would suggest an additional figure, or make reference to the rest of figure 1, which is colour coded to match the schematic in 1a.

The clarity of this description is also diminished by use of KtrAB-specific jargon, e.g. “D3M2”, which is not shown on a figure.

Author response: We have added the topology maps of KtrB from (a) ATP-BsKtrAB and (b) ADP-BsKtrAB. BsKtrB is composed of four structurally similar domains labeled as D1 to D4, and each domain contains three α helices (M1, pore and M2 helices). The “D3M2” helix refers to the M2 helix of domain D3, which is further explained in the figure legend. The functionally important residues G87, F91, T310 and R417 are shown as circles with one letter code of amino acids. This figure has been added as Supplementary Fig. 1.

2. Line 84.

The authors make reference to Slo2.1 and Slo2.2 both here and in the Discussion. While the text indicates a functional link between these proteins and KtrAB (i.e. Na^+ -activated K^+ flux), the presence of this comparison also suggests that they are both structurally and mechanistically linked. However, in my understanding that the only shared domain between these 2 proteins is the RCK domain. I think it would be useful to clarify the text in the introduction to spell out the similarities between these systems and how understanding KtrAB activation could inform on Slo2.1 and 2.2 mechanism.

Author response: Indeed, Slo2.1 and Slo2.2 (K_{Na}) channels are homotetrameric K^+ channels covalently linked to the RCK domain, which is not structurally identical to BsKtrAB complex. However, one previous study using electrophysiological approach demonstrated that the Na^+ -dependent activation of K_{Na} channel requires NAD^+ , which is also a nucleotide (Tamsett et al. J Neurosci 2009). The authors proposed a putative NAD^+ binding site at the RCK domain, which is very similar the DNA-binding site of BsKtrA. Due to the functional and structural similarity, we hope that the synergistic activation of ATP and Na^+ on the basis of our findings in BsKtrAB can provide mechanistic insight into K_{Na} channels. We have revised the introduction to

emphasize the similarity in terms of nucleotide-dependent Na⁺ activation between KNa and KtrAB (Line 83-91).

3. I think the first section (p.5-9), in which a variety of structural approaches (and ICP-MS) are used to convincingly argue that the mystery density in the structures is a Na⁺ ion, is clearly described and well laid out.

Author response: We thank the reviewer's positive comments.

4. The authors use a urea-mediated unfolding assay to compare the stabilizing effects of Na⁺ and K⁺. I am a bit confused about the rationale behind the experimental design. In the Na⁺ titration (3a), the protein is in buffer containing 150 mM K⁺, and in the K⁺ titration the protein is in buffer containing 150 mM Na⁺. If the goal of this assay was to see if Na⁺ and K⁺ can stabilise the protein, would it not have been better to have the protein in buffer containing a functionally inert cation, like choline? In the presence of the 150 mM Na⁺, the protein looks to be "fully stabilized" by Na⁺ binding, so how did the authors expect to see any further stabilization by addition of K⁺?

Author response: In the manuscript, the crystallographic studies of ATP-BsKtrA using TI+ anomalous scattering leaves an ambiguity of Na⁺ and K⁺ in the binding site. Therefore, we performed urea unfolding assay of ATP-BsKtrA in K⁺ or Na⁺ buffer with the titration of Na⁺ or K⁺, respectively. It is noted that the urea unfolding concentration (C_m) of ATP-BsKtrA in K⁺ buffer before Na⁺ titration is low, approximately 2M (Fig. 3a) compared to the C_m (between 4~5 M) of ATP-BsKtrA in Na⁺ buffer before any K⁺ titration (Fig. 3b), indicating ATP-BsKtrA is much more unstable in the presence of K⁺ buffer than Na⁺ buffer. Based on the structural model, the instability of BsKtrA upon ATP binding is due to the negative charge repulsion between the γ -phosphates of the two adjacent ATP molecules, and a favorable cation binding at the site would improve the protein stability. The urea unfolding results reveal that Na⁺ titration specifically enhances the stability of ATP-BsKtrA when the protein is in K⁺ buffer, suggesting Na⁺ favors the binding site. In contrast, when ATP-BsKtrA is in Na⁺ buffer, in which the protein has attained a stable condition, titration of K⁺ cannot cause any difference in C_m, indicating K⁺ cannot compete with Na⁺ in the binding site. In the following stability assays such as DSF (Fig. 3c) and half-life characterization (Fig. 3d), we used choline chloride in the protein buffer to study the stabilization effect of individual cations. One can observe the significant effect of stabilization in ATP-BsKtrA only in the presence of Na⁺. We have clarified the experimental details in the text (Line 214-219 and 234-236).

5. The authors use a SYPRO-based DSF assay to monitor the stabilizing effects of Na⁺ and K⁺, which convincingly shows that there is a dose-dependent stabilization by Na⁺, but not by K⁺. The authors then use the ΔT_m values to plot a binding curve to derive a K_d. While this analysis looks tempting, these data cannot reliably be used to derive affinities because the ΔT_m data are collected at different temperatures, and it cannot be assumed that the binding affinity is constant at different temperatures. There are ways of doing it (see Bai et al. Scientific Reports 9, article number 2650 (2019)), but this is not the approach used here. The qualitative conclusions drawn do not need this additional analysis anyway.

Author response: As reviewer #1 suggested, we have changed constant as the apparent K_d, K_{dapp}, to describe the affinity.

6. Line 242.

The authors state they have “solid evidence that Na⁺ plays a critical role in binding of ATP to BsKtrA.” I would argue the complete opposite based on the ITC data provided. The authors only observed a 3.2-fold increase in the K_d for ATP in the presence of 200 mM Na⁺. These data clearly demonstrate that KtrA is perfectly capable of binding ATP at a reasonable high affinity in the absence of Na⁺.

Author response: We totally agree with the reviewer’s argument. Indeed, 3.2-fold increase of K_d in the ITC results did not provide a strong support that Na⁺ enhances ATP binding. Actually, BsKtrA does not necessarily require Na⁺ to interact with ATP from the ITC data (K_d = 5.5 ± 1.0 μM). However, upon binding of ATP to BsKtrA, Na⁺ can neutralize the negatively charged repulsion between two ATP molecules and stabilize the ATP-bound state, probably accounting for the 3-fold increase of affinity in ITC results. We have rephrased the statement (Line 248-252 and 260-261).

7. Line 249.

See previous comment about deriving K_ds from DSF.

Author response: We have changed the term as K_{dapp} in the revised manuscript.

8. Line 272-279 The K⁺ flux assay.

The data presented by the authors to this point in the manuscript suggests that K⁺ does not interact with KtrAB, so would not be expected to activate the channel, and yet, the authors observe enhanced K⁺ flux in the presence of K⁺ (a substantial 20% enhancement). This needs to be explained and discussed.

Author response: As the ITC results mentioned above, BsKtrA can interact with ATP in the presence of K⁺ (no Na⁺). However, at this ligand binding state the octameric ring is probably not stably maintained at activated conformation (squared-shape octameric ring, Fig. 2a) because of the repulsion of ATP molecules. At this conformational stage, BsKtrA octameric ring cannot provide a substantial steric hinderance to facilitate the helix hairpin conformation of BsKtrB D1M2b helix and thus it only partially activates K⁺ flux activity of BsKtrB. This is probably the reason accounting for the observed 20% enhancement of K⁺. Upon addition of Na⁺, it can neutralize the negative charges between the two ATP molecules and enhance the stability of ATP-BsKtrA octameric ring, therefore stabilizing the assembly with BsKtrB at activated conformation. The original sentence “in the presence of K⁺” means in K⁺ buffer, indicating “the absence of Na⁺”. This is misleading and we have rephrased the sentence accordingly (Line 299).

9. From this assay, the authors observe flux activity under all conditions tested, so this is not really ATP/Na⁺-based activation, it is only activity enhancement, and not really much enhancement either – the presence of Na⁺ and ATP has less than a 2-fold stimulation of activity. Do the authors find this surprising? Is this amount of stimulation physiologically significant?

Author response: The expression level of KtrAB in *B. subtilis* can be regulated by the intracellular [K⁺]. The expression of KtrAB and other K⁺ uptake transporters are downregulated while the intracellular is high through the second messenger, c-di-AMP (Gundlach et al. J Biol Chem 2019). The down regulation is achieved by binding of c-di-AMP to *ydaO* riboswitch

(Block et al. J. Bacteriol. 2010) preventing the translation of KtrAB. Therefore, the Na⁺ dependency of K⁺ uptake may not be significant as the protein expression level of KtrAB is low. While K⁺ is depleted, c-di-AMP synthesis is inhibited and this condition will activate the expression of KtrAB, which would magnify the effect Na⁺ in activation of K⁺ uptake. Although we only observed a less than 2-fold stimulation using in vitro assays, we believe, in bacterial cells, while the expression of KtrAB is elevated under low [K⁺], the activation effect of Na⁺ would be more significant.

10. An important control missing from here is the amount of flux in protein-free liposomes (I note that this control is present in figure 4f, but those results also have some marked differences to these data – see below).

Author response: We have included the control data using protein-free liposomes in the for Fig. 3f (Fig. 4b in the revised manuscript).

11. Figure 4f. The authors should indicate in the figure legend or figure what the black and white arrows indicate (also needed in Figure 3f).

Please could the authors explain why this activity assay only reaches 0.4 before plateauing, whereas the previous data in Figure 3f reached effectively zero. This behaviour is quite different, but 2 of the conditions are the same.

Author response: Thank you for highlighting the confusion between Fig. 3f (Fig. 4b in the revised manuscript) and Fig. 4f (changed to Fig. 5f in the revised manuscript). In the previous manuscript, we have used different normalization procedures in these two figures. In Fig. 3f (changed to Fig. 4b), the fluorescence intensity at 499 second (before adding Valinomycin) was set to be 0. While in Fig. 4f (changed to Fig. 5f), the lowest fluorescence intensity point (after adding Valinomycin) between 500 to 600 second was set to be 0. This explains why the two figures show different normalized level while reaching the plateau at 500 second (0 in Fig.3f, 0.4 in Fig.4f). We have now standardized the normalization process in all K⁺ flux assays as illustrated in Supplementary Fig. 15 and explained in Methods. We also included the trace of the negative control (protein-free liposome) for Fig. 3f (Fig. 4b in the revised manuscript) as requested. Additionally, the black (adding CCCP) and white (adding valinomycin) arrows are now indicated in all figures and described in the figure legends of all K⁺ flux assays.

Reviewer #1 (Remarks to the Author):

I appreciate the authors' care and thoroughness in responding to my critiques. I only have minor textual/figure suggestions – after these are addressed, I recommend this manuscript for publication.

- First, it would be more appropriate to round resolutions to tenth decimals rather than hundredths throughout the text; as in, 2.8 Å instead of 2.82 Å. The extra significant digit is unlikely to be informative at these ranges of resolutions. Apologies for not requesting this in the first round of review.

- Second, to further clarify my previous major comment #6: though I appreciate the authors' efforts and explanation regarding their hypothesis of Na⁺-dependent complex assembly, it still stands that the thermodynamics of assembly may be quite different in detergent vs lipid bilayers. In other words, in the lipid bilayer environment, perhaps Na⁺ stabilizes but is not as strictly required to induce the open state – this would also be consistent with the other reviewer's comments #8-9 and the presented complex formation experiment. This does not affect the value of the excellent work presented, and I do not think additional experiments need to be performed. I simply request that the authors briefly acknowledge this possibility in the manuscript/discussion, and make the distinction between biophysical observations in detergents, and more physiological possibilities in membrane environments.

- Typo (line 269): E1125Q

- Lines 248-252: Again, these statements should be removed – “moderate increase of ATP binding affinity”, and “affinity can be marginally enhanced in the presence of Na⁺.” Without statistical analysis, these claims are unsubstantiated. The proper way to do this would be t-tests using geometric means and standard deviations since these are log-normally distributed data (a common mistake is using arithmetic means/SDs to infer significance between binding affinities). Furthermore, I know that there can be substantial imprecision in modeling the type of binding data displayed in SF13. Anyway, I do not think these statements are important for the paper – so simply removing the statements would be easiest.

- Regarding the K_{dapp} inferred from the T_m data: to further ensure clarity, the phrases “binding affinity” and “dissociation constant” should always be preceded by “apparent”, as in “apparent binding affinity”.

- The other reviewer made an excellent observation regarding the urea-mediated unfolding assay (Point #4). The authors' explanation makes some sense as to the motivation of the experiment, but in the context of the figure it is confusing – in my first review I had completely misinterpreted the purpose of the experiment. If the objective is to rationalize TI⁺ anomalous scattering, these figures should be much better explained in the figure panel. It should be explicitly stated in the figure panel that there is constant 150 mM Na⁺ and K⁺ for the K⁺ and Na⁺ titrations, respectively (inset schematics would be ideal). Furthermore, in the figure legend and text, these should be referred to as competition experiments.

- Alongside the previous point, I think confusion could be avoided if 3a and 3b were presented alongside the TI⁺ data in Figure 2 (Fig 2e-g) or moved to a supplement. I will leave this decision up to the authors.

Reviewer #2 (Remarks to the Author):

The authors have sufficiently addressed my comments.

Reviewer #1 (Remarks to the Author):

I appreciate the authors' care and thoroughness in responding to my critiques. I only have minor textual/figure suggestions – after these are addressed, I recommend this manuscript for publication.

Author response: We appreciate the reviewer's kind suggestion. We have revised the manuscript as requested and highlighted in yellow. In this revision we also noticed some typos, misspellings, missing words and affiliations. We have also amended and highlighted them in cyan.

- First, it would be more appropriate to round resolutions to tenth decimals rather than hundredths throughout the text; as in, 2.8 Å instead of 2.82 Å. The extra significant digit is unlikely to be informative at these ranges of resolutions. Apologies for not requesting this in the first round of review.

Author response: We have changed the resolutions to the tenth decimals in the entire manuscript.

- Second, to further clarify my previous major comment #6: though I appreciate the authors' efforts and explanation regarding their hypothesis of Na⁺-dependent complex assembly, it still stands that the thermodynamics of assembly may be quite different in detergent vs lipid bilayers. In other words, in the lipid bilayer environment, perhaps Na⁺ stabilizes but is not as strictly required to induce the open state – this would also be consistent with the other reviewer's comments #8-9 and the presented complex formation experiment. This does not affect the value of the excellent work presented, and I do not think additional experiments need to be performed. I simply request that the authors briefly acknowledge this possibility in the manuscript/discussion, and make the distinction between biophysical observations in detergents, and more physiological possibilities in membrane environments.

Author response: We have addressed this concern in Discussion (lines 437-440).

- Typo (line 269): E1125Q

Author response: We thank the reviewer's careful inspection. We have amended the typo (line 267).

- Lines 248-252: Again, these statements should be removed – “moderate increase of ATP binding affinity”, and “affinity can be marginally enhanced in the presence of Na⁺.” Without statistical analysis, these claims are unsubstantiated. The proper way to do this would be t-tests using geometric means and standard deviations since these are log-normally distributed data (a common mistake is using arithmetic means/SDs to infer significance between binding affinities). Furthermore, I know that there can be substantial imprecision in modeling the type of binding data displayed in SF13. Anyway, I do not think these statements are important for the paper – so simply removing the statements would be easiest.

Author response: We have crossed out the statements (lines 245-249).

- Regarding the K_d inferred from the T_m data: to further ensure clarity, the phrases “binding affinity” and “dissociation constant” should always be preceded by “apparent”, as in “apparent binding affinity”.

Author response: We have revised these terms in text and figure legends (lines 223, 234, 265 and 738).

- The other reviewer made an excellent observation regarding the urea-mediated unfolding assay (Point #4). The authors' explanation makes some sense as to the motivation of the experiment, but in the context of the figure it is confusing – in my first review I had completely misinterpreted the purpose of the experiment. If the objective is to rationalize Tl⁺ anomalous scattering, these figures should be much better explained in the figure panel. It should be explicitly stated in the figure panel that there is constant 150 mM Na⁺ and K⁺ for the K⁺ and Na⁺ titrations, respectively (inset schematics would be ideal). Furthermore, in the figure legend and text, these should be referred to as competition experiments.

Author response: We apologize for the confusion. We have rephrased the text (lines 206-215) and figure legend (line 218). We also added inset schematics in Fig 3a,b.

- Alongside the previous point, I think confusion could be avoided if 3a and 3b were presented alongside the Tl⁺ data in Figure 2 (Fig 2e-g) or moved to a supplement. I will leave this decision up to the authors.

Author response: We thank the reviewer's suggestion, but we decided to keep the current figure arrangements. The information revealed from Fig. 2e-g is Na⁺ can outcompete Tl⁺ in the binding site but K⁺ cannot. However, the urea unfolding assays (Fig. 3 and Supplementary Fig. 10) demonstrated that Na⁺ and ATP can synergistically enhance protein stability of BsKtrA. We think the two results provide mechanistic insights from different perspectives. We therefore decided to leave it as it is.

Reviewer #2 (Remarks to the Author):

The authors have sufficiently addressed my comments.